# On-Policy Model Error Suffices: An Invariant-Measure Return-Gap Bound for Model-Based Reinforcement Learning

## Abstract

We study the discounted return gap between a fixed policy evaluated on a true dynamical system and on a learned closed-loop model. Lipschitz-based bounds in the model-based reinforcement learning literature control this gap by the *supremum* of the one-step model error over the state space, amplified by the global closed-loop Lipschitz constant; this is pessimistic for systems whose closed-loop trajectories concentrate on a low-dimensional attractor. We prove a return-gap bound whose dominant term is the one-step model error *averaged under the invariant measure of the true closed loop*, amplified by a trajectory-localized linearised contraction rate, plus geometrically-decaying transients. The bound recovers the classical sup-norm bound as a limiting case; its leading term is strictly smaller whenever the invariant-measure-averaged error $\bar{\epsilon}_\mu$ is strictly below the global supremum error $\epsilon_0$, as occurs when large model errors lie off the closed-loop attractor. We exhibit a regime in which this distinction is qualitative: the classical bound is infinite while ours is finite. As a consequence, the empirical on-policy mean-squared error minimized by modern world-model algorithms upper-bounds (up to a square-root and a finite-sample concentration term) the return-gap-controlling quantity, giving the training objective an explicit return-gap interpretation. We extend the result to stochastic dynamics via a Wasserstein-1 coupling, and prove a matching bound on the Wasserstein distance between the true and learned-model invariant measures.

## 1 Introduction

When an agent plans with a learned model of its environment, a basic question is how wrong the resulting evaluation can be when the model is wrong. If the learned dynamics differ from the true dynamics by some amount, how far can the predicted return of a fixed policy diverge from its true return? The classical answer, going back to the simulation lemma and its Lipschitz refinements, charges the worst case: It measures model error by its largest value anywhere in the state space and amplifies that error by how fast nearby trajectories can spread apart. For systems that are being actively stabilized, a robot holding its balance, a vehicle tracking a path, a controller regulating a setpoint, this worst-case accounting is badly pessimistic. The closed-loop trajectories visit only a small, stable region of the state space, so a model that is wildly inaccurate where the system never goes is penalized exactly as heavily as one that is wrong where it matters. This paper makes that intuition precise: For fixed-policy evaluation under a contracting closed loop, the return gap is controlled by the model's *average* error over the states the system actually visits, not by its worst-case error.

Many deployments of learned dynamics models, e.g., off-policy policy evaluation, sim-to-real rollout prediction, model-based value expansion, reliability verification of a pre-trained policy, share a common structure: a *fixed* policy is applied to both a true system $F$ and a learned model $\hat{F}$, and the question is how much $J(\hat{F})$ measured on the learned model can diverge from $J(F)$ on the true system. This is the setting of off-policy evaluation, see, e.g., (Voloshin et al., 2021; Thomas & Brunskill, 2016; Jiang & Li, 2016) and of the evaluation step of many MBRL algorithms, e.g., (Feinberg et al., 2018; Buckman et al., 2018; Hafner et al., 2020). It is not the full policy-optimization problem, in which the policy is itself chosen through the learned model

(Janner et al., 2019; Xu et al., 2022); we leave the policy-optimization extension to future work. The classical bound (Asadi et al., 2018; Gelada et al., 2019; Zhang et al., 2021a), controls $\left|J(F) - J(\hat{F})\right|$ by the supremum of one-step model error $\epsilon_0 := \sup_\Omega \left\|F - \hat{F}\right\|$, amplified by the global closed-loop Lipschitz constant. This framing is structurally pessimistic in two ways that matter for continuous-state learned dynamics: (i) neural world models exhibit arbitrarily large errors in states the deployed closed loop never visits, so $\epsilon_0$ is dominated by off-support regions; and (ii) global Lipschitz constants on continuous-state nonlinear systems routinely exceed one (Lambert et al., 2020; Suh et al., 2022), making the bound vacuous precisely where a practitioner wants it.

**Main focus.** In this work, we bound $\left|J(F) - J(\hat{F})\right|$ for fixed-policy evaluation under closed-loop contraction of the true dynamics, in a form that separates the classical sup-norm bound into three structurally distinct terms, each individually relaxable. Under closed-loop contraction and Wasserstein-1 mixing, our main theorem (Theorem 1, Section 4) controls the return gap by the one-step model error *averaged under the invariant measure $\mu$ of the true closed loop*, with a prefactor determined by a linearised contraction rate along trajectory-error segments and smoothness required only within a localized tube. A direct consequence, stated alongside the theorem, is that the empirical on-policy mean-squared error $\hat{\mathcal{L}}_N$ estimates $\mathbb{E}_\mu \left\|\hat{F} - F\right\|^2$, whose square root upper-bounds the dominant error term by Cauchy–Schwarz; the on-policy MSE objective therefore admits a return-gap interpretation (Corollary 1), conditional on contraction and on-policy sampling. Section 5 then tests the bound numerically: Each of the three relaxations can be strict in natural conditions, and our experiments isolate a setting in which all three are simultaneously active (Figure 2). We state the deterministic result in the body because it exposes the geometric mechanism most directly; the stochastic-kernel extension, where the invariant-measure framing is most natural and $\mu$ is typically non-degenerate under process noise, is derived in Section D, and complete proofs of all results appear in the appendix, with only proof sketches in the body.

**Extensions and experiments.** We additionally prove a matching bound on the $W_1$ distance between the true and learned-model invariant measures (Theorem 3): Under the same contraction and Lipschitz-error conditions, the quantity that controls the return gap also controls the invariant-measure shift between $F$ and $\hat{F}$. The main theorem extends to limit-cycle systems via transverse contraction (Section G). The body contains four numerical tests of the bound: an analytical distribution-shift experiment on a 2D linear system; a nonlinear system on which the classical bound diverges while ours remains tight; finite-sample LQR identification, which recovers the $\mathcal{O}(1/N)$ rate of Mania et al. (2019) with a constant approximately $200\times$ smaller; and a designed system on which the classical bound is infinite at every perturbation magnitude while our bound remains within an order of magnitude of the observed return gap. Two further tests in the appendix, a neural-network world model on a stabilized pendulum and the Van der Pol limit cycle, illustrate the same mechanism in a learned model and a non-equilibrium attractor. A final appendix experiment varies the state dimension up to $d = 100$ and confirms that the invariant-measure-averaged error remains estimable at a dimension-independent rate, so the bound stays tractable in high dimensions (Section F.6).

## 2 Related work

**Lipschitz MBRL.** The closest prior work is Asadi et al. (2018), who prove that under a composed closed-loop Lipschitz constant $K < 1$ and uniform one-step error $\epsilon_0 := \sup_\Omega \left\|F - \hat{F}\right\|$, the value gap is bounded by $\gamma L_r \epsilon_0 / [(1 - \gamma)(1 - K)]$. Extensions of this framework to stochastic models (Gelada et al., 2019; Zhang et al., 2021a), bisimulation metrics (Ferns et al., 2004; Castro, 2020), and dual formulations (Tessler et al., 2019) preserve the worst-case supremum framing. We recover this bound as the $C^0$ corollary of our main theorem (Corollary 3) and strengthen it in three structurally distinct ways: The supremum error $\epsilon_0$ is replaced by the invariant-measure expectation $\bar{\epsilon}_\mu \leq \epsilon_0$; the global Lipschitz constant $K$ is replaced by a linearized Jacobian rate $\bar{\rho} \leq K$; and the uniform $C^1$ requirement is replaced by a tube-localized one. Each relaxation can be strict under natural conditions. Each relaxation weakens a worst-case element of the classical bound and can be strict in natural settings. Figure 2 gives a concrete example in which the invariant-measure error,

trajectory-localized contraction rate, and tube-localized regularity all improve the bound while the classical sup-norm bound is vacuous. At short horizons, however, the transient terms may dominate the leading invariant-measure term.

**Stability-based imitation learning.** The closest *mechanism* to ours appears in imitation learning: TaSIL Pfrommer et al. (2022) and Tu et al. (2022) prove $\tilde{O}(1/n)$ imitation gaps under $\delta$-input-to-state stability of the expert, using a Taylor-matching between learner and expert policies. The shared principle is that closed-loop stability prevents local approximation errors from accumulating indefinitely. However, the objects being compared are different: TaSIL compares policies on fixed true dynamics, whereas we compare true and learned dynamics under a fixed policy. The answer is that the *objects* are orthogonal and the *measure-transfer step is specific to the dynamics-learning setting.* TaSIL bounds the gap between an expert policy $\pi_{\text{expert}}$ and a learned policy $\hat{\pi}$ when both are deployed on *fixed true dynamics*; the error quantity is the policy-parameter deviation $\|\hat{\pi} - \pi_{\text{expert}}\|$ in a Taylor-matching metric, and the stabilizing assumption is $\delta$-ISS of the expert's closed loop. We bound the gap between true dynamics $F$ and learned dynamics $\hat{F}$ when both are driven by a *fixed policy*; the error quantity is the one-step *dynamics* error $\left\| F - \hat{F} \right\|$, averaged under the invariant measure of the deployed closed loop, and the stabilizing assumption is closed-loop contraction of the true dynamics. Neither bound is an instance of the other. In particular, TaSIL has no analog of our Kantorovich–Rubinstein measure-transfer step (Lemma 3): It imitates a fixed expert, so there is no "true-system invariant measure" in their setting distinct from the expert's rollout distribution. Our measure-localization is therefore new information, not a re-derivation of Pfrommer et al. (2022) in different notation.

**Other related mechanisms.** Suh et al. (2022) identify chaos and nonsmoothness as failure modes of first-order differentiable-simulation gradients; our contraction assumption excludes their chaos condition, and the two analyses are complementary. Grimm et al. (2020; 2021) match the learned model on the Bellman operator rather than on one-step prediction, a distinct task-weighting philosophy (value-equivalence vs. invariant-measure-equivalence). Empirical work on world-model smoothness regularization (Singh et al., 2021; Georgiev et al., 2025; Pfrommer et al., 2021) is explained quantitatively by our bound: the Lipschitz constant of the one-step error controls both transient terms. Horizon-free regret bounds for tabular and linear MDPs (Zhang et al., 2021b; Wang et al., 2020; 2025) achieve horizon-independence via variance reduction; our $(1 - \bar{\rho})^{-1}$ plays the analogous role for deterministic continuous-state dynamics. Empirically, Wissmann et al. (2024) observe exactly the phenomenon our bound formalizes: For fixed-policy closed-loop evaluation, long model-based rollouts yield far better value estimates than worst-case one-step-error reasoning predicts, so the pessimism of the sup-norm accounting is an artifact of the analysis rather than of the rollouts. On the algorithmic side, Talvitie (2014) and the follow-up self-correcting-model line (Talvitie, 2017) address the consequences of one-step model error for rollout stability by regularizing the model toward self-consistency; our analysis is complementary, characterizing when the unregularized rollout error is already benign because the closed loop contracts.

**Finite-sample system identification.** Dean et al. (2020) and Simchowitz & Foster (2020) provide finite-sample bounds on $\epsilon_0$ for LQR system identification, and Mania et al. (2019) prove certainty-equivalence suboptimality for LQR control. These results supply the data-to-model-error half of the sample-complexity-to-return-gap pipeline; our Theorem 1 supplies the model-error-to-return-gap half for the nonlinear setting under contraction.

**Distinctions from close relatives.** *Compare with TaSIL* (Pfrommer et al., 2022): TaSIL bounds an imitation gap under expert $\delta$-ISS via Taylor-matching on a *learned policy*; we bound an evaluation gap under true-system contraction via invariant-measure averaging of a *learned dynamics model.* The stabilizing assumption, the bounded object, and the Kantorovich–Rubinstein measure-transfer step are distinct.

*Compare with the simulation lemma* (Kakade & Langford, 2002; Janner et al., 2019): Kakade–Langford and MBRL descendants bound the gap by expected error under the *learned model's* rollout distribution, requiring a secondary rollout-shift term. We bound the gap directly under the *true* system's invariant measure, so the rollout-shift step is unnecessary.

*Compare with value-aware model learning* (Grimm et al., 2020; 2021): value-equivalence matches on the Bellman operator (requires known value function); we use only dynamics error, the quantity on-policy world-model training optimizes.

## 3 Problem setup

We evaluate a fixed policy $\pi$ on true dynamics $f$ versus its learned approximation $\hat{f}$. Writing $F(x) := f(x, \pi(x))$ and $\hat{F}(x) := \hat{f}(x, \pi(x))$ for the closed-loop maps, the problem reduces to comparing two deterministic iterated systems $x_{t+1} = F(x_t)$ and $\hat{x}_{t+1} = \hat{F}(\hat{x}_t)$ on a compact forward-invariant region $\Omega \subseteq \mathcal{X}$. For bounded reward $r : \mathcal{X} \to \mathbb{R}$ and discount $\gamma \in (0, 1)$,

$$J(F) := \sum_{t=0}^{\infty} \gamma^t r(x_t), \qquad J(\hat{F}) := \sum_{t=0}^{\infty} \gamma^t r(\hat{x}_t), \qquad x_0 = \hat{x}_0 \in \Omega. \tag{1}$$

Our goal is a tight bound on $|J(F) - J(\hat{F})|$ in terms of interpretable quantities that a practitioner can measure or control: the invariant measure of the deployed closed loop, the contraction rate along its trajectories, and the one-step error of the learned model restricted to the states the policy actually visits.

This fixed-policy setting covers model-based off-policy evaluation (Voloshin et al., 2021), sim-to-real policy deployment, and the inner loop of value-expansion and model-based policy-optimization methods where the policy is held fixed for an optimization step (Feinberg et al., 2018; Janner et al., 2019). It is the natural unit of analysis: policy optimization with a learned model can be decomposed into fixed-policy evaluation plus a policy-improvement bias, and the fixed-policy term is what classical Lipschitz-MBRL bounds address.

### 3.1 Assumptions

We state the five assumptions used throughout and briefly justify each.

**Assumption 1** (Bounded operating region with anchor). There exist $x_\star \in \Omega$ and $R \geq 0$ such that $\sup_{x \in \Omega} \|x - x_\star\| \leq R$.

The anchor $x_\star$ represents the closed-loop equilibrium or reference point that the policy is designed to stabilize objectives such as the upright pose for an inverted pendulum, the reference trajectory for a tracking controller, or the setpoint for a regulator. It is the natural point at which to localize error.

**Assumption 2** (Closed-loop contraction on $\Omega$). $F$ is Lipschitz on $\Omega$ with constant $\rho \in [0, 1)$: $\|F(x) - F(y)\| \leq \rho \|x - y\|$ for all $x, y \in \Omega$.

This is the substantive assumption of the paper. It requires the true closed-loop map induced by the deployed policy to be contractive: any two trajectories initialized in $\Omega$ move closer together in Euclidean distance at each step. This holds for stabilizing policies on linear systems with $\|A + BK\|_2 < 1$, for nonlinear systems around a stable equilibrium where the Jacobian of the closed loop is a contraction, and more generally for systems equipped with a problem-adapted contraction metric (Lohmiller & Slotine, 1998; Manchester & Slotine, 2017). It excludes chaotic conditions, limit-cycle locomotion analyzed in ambient coordinates (though transverse contraction recovers a variant, see Section 6), and contact-rich dynamics with Jacobian discontinuities.

**Assumption 3** (Lipschitz reward). $r$ is $L_r$-Lipschitz on $\mathcal{X}$: $|r(x) - r(y)| \leq L_r \|x - y\|$.

Standard: all quadratic, bounded-differentiable, and bounded-Lipschitz rewards satisfy this.

**Assumption 4** (Forward invariance and convexity). For every $x \in \Omega$, both $F(x) \in \Omega$ and $\hat{F}(x) \in \Omega$, and $\Omega$ is convex. Convexity is required because Lemmas 1 and 2 integrate $\nabla F$ and $\nabla \hat{F}$ along straight-line segments between trajectory states; without convexity, those segments may exit $\Omega$.

The two closed loops keep trajectories in the operating region. When the true closed loop contracts and $\hat{F}$ is close to $F$, this holds for $\Omega$ chosen as a Euclidean ball around $x_\star$, which is convex by construction. We emphasize that forward invariance is required for the *learned* model $\hat{F}$, not only for the true system $F$:

The tube $C^1$ bound of Assumption 5 controls the Jacobian error only along segments contained in $\Omega$, and without forward invariance of $\hat{F}$ the learned rollout could leave the region where this control holds. The assumption is mild in practice. For a strictly contracting true closed loop, a sublevel set of the associated Lyapunov function is forward-invariant for $F$; when $\hat{F}$ is sufficiently close to $F$, a slightly enlarged sublevel set is forward-invariant for $\hat{F}$ as well, so both maps keep their rollouts inside $\Omega$. This is the situation in all of our experiments, where $\Omega$ is a Euclidean ball and both rollouts remain interior to it.

**Assumption 5** (Anchored local $C^1$ model error). There exists a connected set $\mathcal{T} \subseteq \Omega$ containing the anchor $x_\star$ and all line segments $[x_\star, \hat{x}_k]$ and $[\hat{x}_k, x_k]$ for all $k \geq 0$ (the segments are themselves contained in $\Omega$ by the convexity of Assumption 4). On this tube $\mathcal{T}$, $F, \hat{F} \in C^1(\mathcal{T}; \mathbb{R}^d)$ with $\left\| F(x_\star) - \hat{F}(x_\star) \right\| \leq \delta$ and $\tilde{\epsilon}_1 := \sup_{\xi \in \mathcal{T}} \left\| \nabla F(\xi) - \nabla \hat{F}(\xi) \right\|_{\text{op}} < \infty$.

This replaces the classical $C^0$ supremum assumption $\sup_\Omega \left\| F - \hat{F} \right\| \leq \epsilon_0$ by two localized quantities: a scalar error $\delta$ at the anchor, and a tube-Jacobian error $\tilde{\epsilon}_1$ restricted to the segments that the proof actually integrates along. The global $C^1$ version ($F, \hat{F} \in C^1(\Omega)$ with $\sup_\Omega \left\| \nabla F - \nabla \hat{F} \right\| \leq \epsilon_1$) is a convenient sufficient condition that implies Assumption 5 with $\tilde{\epsilon}_1 \leq \epsilon_1$; we use the tube-localized form throughout since this is the form the theorem actually requires. The two imply a $C^0$ bound $\epsilon_0 \leq \delta + R\tilde{\epsilon}_1$ via the fundamental theorem of calculus (Lemma 2), but the separation is what enables our tube-localization argument: derivative error along the segments joining an anchor to the current state is what matters, not function-value error on all of $\Omega$. In MBRL practice, $\delta$ is a single scalar constraint that on-policy training enforces trivially, while $\tilde{\epsilon}_1$ is bounded by Jacobian-matching losses that methods such as PWM (Georgiev et al., 2025) and contraction-regularized models (Singh et al., 2021) already optimize.

## 3.2 The invariant measure

The novelty of our analysis turns on a further structural object: the invariant measure $\mu$ of the true closed loop $F$. Under Assumptions 2 and 4, $F$ admits a unique Borel probability measure $\mu$ on $\Omega$ satisfying $F_\#\mu = \mu$. *In the strictly contracting deterministic setting, $\mu$ collapses to a point mass at the fixed point $x_\star$, so $\bar{\epsilon}_\mu = \left\| F(x_\star) - \hat{F}(x_\star) \right\| = \delta$ reduces to the anchor model error.* This means the deterministic invariant-measure framing is, in the literal-equilibrium setting, equivalent to a local-attractor analysis at $x_\star$. The full invariant-measure interpretation only becomes structurally rich in three settings: *(i)* stochastic dynamics with process noise (Section D), where $\mu$ is typically non-degenerate; *(ii)* systems whose attractor is a limit cycle rather than a fixed point (Section G), where the invariant measure is supported on the cycle; *(iii)* piecewise contraction on separate basins (where global contraction Assumption 2 holds on each basin separately rather than on $\Omega$ globally), and $\mu$ reflects the basin distribution. Settings *(i)* and *(ii)* are within the scope of this paper; *(iii)* is a natural extension we do not develop here. The object of interest is the *expected one-step model error* under $\mu$:

$$\bar{\epsilon}_\mu := \mathbb{E}_{x \sim \mu} \left\| F(x) - \hat{F}(x) \right\|. \tag{2}$$

This quantity is strictly smaller than the supremum error $\epsilon_0$ whenever $\mu$ is not uniform on $\Omega$, which is the generic case for a stabilized control system whose trajectories concentrate near the equilibrium. The main theorem of Section 4 bounds the return gap in terms of $\bar{\epsilon}_\mu$ rather than $\epsilon_0$. The magnitude of the improvement $\epsilon_0/\bar{\epsilon}_\mu$ is the theoretical payoff of *measure-localization*: it quantifies how much tighter a distribution-aware bound is than the classical worst-case one.

The on-policy empirical MSE $\hat{\mathcal{L}}_N = \frac{1}{N} \sum_i \left\| F(x_i) - \hat{F}_\theta(x_i) \right\|^2$ minimized by world-model training estimates $\mathbb{E}_\mu \left\| F - \hat{F} \right\|^2$, whose square root upper-bounds $\bar{\epsilon}_\mu$ by Cauchy–Schwarz; Corollary 1 makes this explicit. The framing extends to stochastic dynamics ($\mu$ typically non-degenerate, $\bar{\epsilon}_\mu = \mathbb{E}_\mu W_1(P, \hat{P})$; see Section D).

## 4 Main results

We first present the bound for deterministic closed-loop dynamics, where the proof's geometry is most transparent: The trajectory error is a deterministic sequence, the Jacobian recursion is a literal product of matrices over trajectory-error segments, and the tube-FTC argument anchors at a fixed equilibrium. Section D develops the stochastic kernel version (Theorem 2), where the invariant-measure interpretation is most natural, i.e., the closed-loop kernel $P$ has a typically non-degenerate stationary measure $\mu$ on $\Omega$ (not necessarily absolutely continuous in pathological cases, but generically so for noisy stabilized systems), and $\bar{\epsilon}_\mu = \mathbb{E}_\mu W_1(P, \hat{P})$ genuinely averages model error over the support of the rollout distribution. The deterministic case in this section is most useful as a clear statement of the geometric mechanism; the stochastic extension is the one most directly relevant to learned world models with process noise. The proof assembles a synchronously-coupled variational recursion, a tube-localized fundamental theorem of calculus, and a Kantorovich–Rubinstein mixing argument converting trajectory suprema to expectations under $\mu$.

### 4.1 The invariant-measure return-gap bound

**Theorem 1** (Invariant-measure return-gap bound). *Assume Assumptions 1 to 5. Suppose additionally that (a) the true closed loop $F$ admits a unique invariant probability measure $\mu$ on $\Omega$ with Wasserstein-1 mixing rate $W_1(\delta_x F^t, \mu) \leq C_{\mathrm{mix}} \alpha^t$ for some $\alpha \in [0, 1)$, and (b) the one-step error $\psi(x) := \left\| F(x) - \hat{F}(x) \right\|$ is $L_\psi$-Lipschitz on $\Omega$. Let $\bar{\rho} := \sup_{t \geq 0} \sup_{\xi \in [\hat{x}_t, x_t]} \|\nabla F(\xi)\|$ denote the linearized contraction rate along the trajectory error segments, $\bar{\epsilon}_\mu := \mathbb{E}_\mu \left\| F - \hat{F} \right\|$ the $\mu$-averaged error, and $\bar{R} := \sup_k \|\hat{x}_k - x_\star\|$ the learned-trajectory radius. Then for any $\gamma \in (0, 1)$ and any $x_0 = \hat{x}_0 \in \Omega$,*

$$
\left| J(F) - J(\hat{F}) \right| \leq \frac{\gamma L_r}{(1-\gamma)(1-\bar{\rho})} \left( \bar{\epsilon}_\mu + \underbrace{\frac{L_\psi C_{\mathrm{mix}} (1-\gamma)}{1 - \gamma \alpha}}_{\text{mixing transient}} + \underbrace{\frac{L_\psi (\delta + \tilde{\epsilon}_1 \bar{R})}{1 - \bar{\rho}}}_{\text{model-drift transient}} \right). \tag{3}
$$

*where $\delta = \left\| F(x_\star) - \hat{F}(x_\star) \right\|$ and $\tilde{\epsilon}_1 := \sup_{\xi \in \mathcal{T}} \left\| \nabla F(\xi) - \nabla \hat{F}(\xi) \right\|$ is the Jacobian error on the tube $\mathcal{T} := \bigcup_{k \geq 0} [x_\star, \hat{x}_k]$.*

For deterministic strict contractions, the Wasserstein-1 mixing condition (a) of Theorem 1 is automatically satisfied with $\mu = \delta_{x_\star}$, so the deterministic bound is, in the literal-equilibrium sense, an anchor-error theorem. The genuinely non-degenerate invariant-measure averaging appears in the stochastic case (Section D) and in the limit-cycle case (Section G).

The bound has three operational components. The dominant term $\bar{\epsilon}_\mu$ is the expected one-step model error under the true closed-loop invariant measure, i.e., the *structural* quantity that the classical sup-norm bound replaces with a worst-case supremum. The mixing transient decays geometrically at a rate $\alpha$ to the true system equilibrium $\mu$; it is controlled by the system and independent of the model. The model-drift transient decays geometrically through the contraction of $F$ and is controlled by the *anchor error* $\delta$ and *tube Jacobian error* $\tilde{\epsilon}_1$, both of which reduce to the anchor $x_\star$ and segments terminating at the trajectory, not to uniform quantities on $\Omega$.

*Remark* 1 ($L_\psi$ is a first-order model-error quantity). For $F, \hat{F} \in C^1(\Omega)$ with $\Omega$ convex, the reverse triangle inequality and the fundamental theorem of calculus give $|\psi(x) - \psi(y)| \leq \left\| (F - \hat{F})(x) - (F - \hat{F})(y) \right\| \leq \sup_{\xi \in \Omega} \left\| \nabla F(\xi) - \nabla \hat{F}(\xi) \right\| \|x - y\|$, hence

$$
L_\psi \leq \sup_\Omega \left\| \nabla F - \nabla \hat{F} \right\|.
$$

Hypothesis (b) of Theorem 1 is therefore a $C^1$ model-quality condition: supremum accuracy alone does not control $L_\psi$ (a small but rapidly oscillating error field has $\epsilon_0 \to 0$ with $L_\psi$ arbitrarily large), whereas Jacobian

accuracy makes $L_\psi$ first order in the model error, and hence makes the model-drift transient in Equation 3, proportional to $L_\psi(\delta + \tilde\epsilon_1 \bar{R})$, second order.

## 4.2 The three technical ingredients

**Lemma 1** (Linearized variational recursion). *Under Assumption 5 with $\Omega$ convex, the error $e_t := x_t - \hat{x}_t$ satisfies*

$$e_{t+1} = A_t\, e_t + \Delta_t, \qquad A_t := \int_0^1 \nabla F(\hat{x}_t + s e_t)\, \mathrm{d}s, \qquad \Delta_t := F(\hat{x}_t) - \hat{F}(\hat{x}_t), \tag{4}$$

*with $e_0 = 0$, $\|A_t\| \le \bar\rho$, and the unrolled bound $\|e_t\| \le \sum_{k=0}^{t-1} \bar\rho^{\,t-1-k}\, \|\Delta_k\|$.*

The key technical move is that $A_t$ is an *averaged* Jacobian along the segment $[\hat{x}_t, x_t]$ rather than a pointwise Jacobian, obtained from the vector-valued fundamental theorem of calculus. This averaged form is what permits the contraction rate to be the trajectory-localized $\bar\rho$ rather than the global Lipschitz constant $\rho$; the inequality $\bar\rho \le \rho$ is strict whenever $\nabla F$ varies on $\Omega$ and trajectories concentrate in a subregion.

**Lemma 2** (Trajectory-tube one-step error). *Under Assumptions 1 and 5,*

$$\|\Delta_k\| \ \le \ \delta + \tilde\epsilon_1\, \|\hat{x}_k - x_\star\|, \qquad \tilde\epsilon_1 := \sup_{\xi \in \mathcal{T}} \left\| \nabla F(\xi) - \nabla \hat{F}(\xi) \right\|. \tag{5}$$

Again, via the fundamental theorem of calculus, this time on the segment $[x_\star, \hat{x}_k]$ joining the anchor to the learned trajectory. The crucial consequence: the Jacobian-error requirement is localized to the tube $\mathcal{T}$, which under closed-loop contraction converges to a neighborhood of $x_\star$ whose radius is $O(\bar\epsilon_\mu/(1-\bar\rho))$, far smaller than $\Omega$ in general.

**Lemma 3** (Measure transfer under mixing). *Under assumption (a) of Theorem 1, for any $L_\psi$-Lipschitz $\psi$,*

$$\left| \psi(x_t) - \mathbb{E}_\mu \psi \right| \ \le \ L_\psi\, C_{\mathrm{mix}}\, \alpha^t. \tag{6}$$

Immediate from Kantorovich–Rubinstein duality: the Wasserstein-1 distance between the pushforward $\delta_{x_0} F^t$ (a point mass at $x_t$) and $\mu$ is at most $C_{\mathrm{mix}} \alpha^t$, and a Lipschitz test function integrates to within the same factor. This is the step that *replaces the supremum by an expectation*, at the cost of a geometrically decaying transient.

## 4.3 Proof sketch of Theorem 1

We outline the proof here; the complete proof, with all constants tracked, is given in Section B.

The proof transforms a quantity we cannot easily control into one we can. By the telescoping argument below, the return gap is governed by the one-step model error accumulated along trajectories, but evaluated as a *supremum over trajectory states*, which is precisely the worst-case object we wish to avoid. The goal is to replace this trajectory supremum by an expectation under the invariant measure $\mu$. The decomposition in Equation 8 helps to split the accumulated one-step error into a model-drift term (how far the learned trajectory departs from the true one before contraction pulls it back), a mixing term (how quickly the trajectory occupancy converges to $\mu$), and the invariant-measure-averaged error $\bar\epsilon_\mu$ itself, which is the quantity we ultimately want. The three lemmas of the previous subsection bound these pieces in turn, and the contraction assumption is what makes the model-drift and mixing terms decay geometrically rather than accumulate.

Apply Lemma 1 to bound $\|e_t\|$, then the Lipschitz-reward Assumption 3 to bound the return gap:

$$\left| J(F) - J(\hat{F}) \right| \le L_r \sum_{t=1}^{\infty} \gamma^t \|e_t\| \le L_r \sum_{t=1}^{\infty} \gamma^t \sum_{k=0}^{t-1} \bar\rho^{\,t-1-k} \|\Delta_k\| = \frac{\gamma L_r}{1 - \gamma\bar\rho} \sum_{k=0}^{\infty} \gamma^k \|\Delta_k\|, \tag{7}$$

where the last step swaps the order of summation (by Tonelli's theorem, which applies since all terms are non-negative). Decompose $\|\Delta_k\| = \psi(\hat{x}_k)$ as

$$\psi(\hat{x}_k) = \underbrace{[\psi(\hat{x}_k) - \psi(x_k)]}_{\text{(I) model-drift}} + \underbrace{[\psi(x_k) - \mathbb{E}_\mu \psi]}_{\text{(II) mixing transient}} + \underbrace{\mathbb{E}_\mu \psi}_{\text{(III)} = \bar{\epsilon}_\mu}. \tag{8}$$

Term (I) is bounded by $L_\psi \|\hat{x}_k - x_k\|$ (Lipschitz of $\psi$) and then by $L_\psi(\delta + \tilde{\epsilon}_1 \bar{R})/(1 - \bar{\rho})$ (Lemma 2 and Lemma 1). Term (II) is $L_\psi C_{\text{mix}} \alpha^k$ (Lemma 3). Term (III) is $\bar{\epsilon}_\mu$. Substituting and evaluating the geometric sums $\sum_{k \geq 0} \gamma^k = 1/(1-\gamma)$, $\sum_{k \geq 0} (\gamma\alpha)^k = 1/(1-\gamma\alpha)$, then using $(1-\gamma\bar{\rho}) \geq (1-\bar{\rho})$ to simplify the prefactor, yields Equation 3. Full details in Section B.

## 4.4 Consequences

Three corollaries follow directly. The first connects the bound to the standard world model training; the second identifies the asymptotic regime; the third recovers the classical sup-norm bound as a special case.

**Corollary 1** (Return-gap control via on-policy training). *Suppose the learned model satisfies $\delta \leq \delta_0$. Let $\{x_i\}_{i=1}^N$ be i.i.d. samples from $\mu$, and let $\mathcal{G} = \{x \mapsto \left\|F(x) - \hat{F}_\theta(x)\right\|^2 : \theta \in \Theta\}$ be the squared-error class with Rademacher complexity Bartlett & Mendelson (2002) $\mathfrak{R}_N(\mathcal{G})$. Define the empirical MSE $\hat{\mathcal{L}}_N := \frac{1}{N} \sum_{i=1}^N \left\|F(x_i) - \hat{F}(x_i)\right\|^2$. With probability at least $1 - \eta$ over the samples,*

$$\bar{\epsilon}_\mu \leq \sqrt{\hat{\mathcal{L}}_N + 2\mathfrak{R}_N(\mathcal{G}) + B^2 \sqrt{\frac{\log(2/\eta)}{2N}}}, \tag{9}$$

*where $B = \sup_{x \in \Omega, \theta \in \Theta} \left\|F(x) - \hat{F}_\theta(x)\right\|$. The square root composition is Jensen's inequality: $\bar{\epsilon}_\mu = \mathbb{E}_\mu \left\|F - \hat{F}\right\| \leq \sqrt{\mathbb{E}_\mu \left\|F - \hat{F}\right\|^2}$. Substituting into Equation 3 bounds the return gap as a function of the empirical MSE plus model-class complexity plus the decaying transients of Theorem 1; full proof in Section C.*

This corollary makes two idealizations: i.i.d. samples (rollouts produce trajectory-correlated samples with a mixing-time variance correction (Mohri & Rostamizadeh, 2010; Kuznetsov & Mohri, 2017)) and a squared-error function class $\mathcal{G}$ admitting finite-sample concentration. The implication: Under closed-loop contraction and on-policy data, the empirical MSE that world-model training minimizes is, up to a square-root and complexity term, an upper bound on the return-gap-controlling quantity $\bar{\epsilon}_\mu$.

**Corollary 2** (Asymptotic regime). *In the regime where the two transient terms in Equation 3 are dominated by the leading error,*

$$\frac{L_\psi C_{\text{mix}}(1-\gamma)}{1-\gamma\alpha} + \frac{L_\psi(\delta + \tilde{\epsilon}_1 \bar{R})}{1-\bar{\rho}} = o(\bar{\epsilon}_\mu), \tag{10}$$

*the bound reduces to*

$$\left|J(F) - J(\hat{F})\right| \leq (1 + o(1)) \frac{\gamma L_r}{(1-\gamma)(1-\bar{\rho})} \bar{\epsilon}_\mu. \tag{11}$$

*The condition Equation 10 requires only the model-error quantities $L_\psi, \delta, \tilde{\epsilon}_1$ to be small; system constants $C_{\text{mix}}, \alpha, \bar{R}$ need not vanish.*

**Corollary 3** ($C^0$ baseline, recovering Asadi et al. (2018)). *Under Assumptions 1 to 4 and the supremum-error assumption $\sup_\Omega \left\|F - \hat{F}\right\| \leq \epsilon_0$,*

$$\left|J(F) - J(\hat{F})\right| \leq \frac{\gamma L_r}{(1-\gamma)(1-\rho)} \epsilon_0. \tag{12}$$

This follows from the scalar recursion $\|e_{t+1}\| \leq \rho \|e_t\| + \epsilon_0$ and is the bound proved by Asadi et al. (2018). It is the special case of Theorem 1 obtained by using the global Lipschitz rate $\rho$ in place of $\bar{\rho}$ and the supremum error $\epsilon_0$ in place of $\bar{\epsilon}_\mu$.

**Three independent improvements.** Comparing Equation 3 to Equation 12: (i) the prefactor is $(1 - \bar{\rho})^{-1} \le (1 - \rho)^{-1}$; (ii) the dominant error quantity is $\bar{\epsilon}_\mu \le \epsilon_0$; (iii) the remaining transient terms depend on the anchor error $\delta$, tube-Jacobian error $\tilde{\epsilon}_1$, and mixing time $C_{\mathrm{mix}}/(1-\alpha)$ rather than on the global supremum error. The leading term gains both improvements multiplicatively (prefactor $1/(1 - \bar{\rho})$ times $\bar{\epsilon}_\mu$); the full bound inherits this gain when the transient terms are controlled. Even when $\bar{\rho} = \rho$ and the transients are non-negligible, the leading term still benefits from the measure-localization factor $\bar{\epsilon}_\mu/\epsilon_0$, typically small for stabilized control systems whose invariant measures concentrate near the equilibrium.

**Corollary 4** (Tube-supremum baseline). *Under Assumptions 1 to 5,*

$$\left| J(F) - J(\hat{F}) \right| \ \le \ \frac{\gamma L_r \left( \delta + \tilde{\epsilon}_1 \bar{R} \right)}{(1 - \gamma\bar{\rho})(1 - \gamma)}. \tag{13}$$

Immediate from Equation 7: bound $\|\Delta_k\| \le \delta + \tilde{\epsilon}_1 \|\hat{x}_k - x_\star\| \le \delta + \tilde{\epsilon}_1 \bar{R}$ by Lemma 2, and sum the geometric series. It requires neither the mixing hypothesis (a) nor the Lipschitz hypothesis (b) of Theorem 1, and it charges the trajectory-tube error envelope, the largest one-step error the learned rollout can meet, at every one of the $1/(1 - \gamma)$ effective steps.

**Where each bound is tighter.** Corollary 3, Corollary 4, and Theorem 1 form a ladder: the classical bound charges the domain supremum $\epsilon_0$; the tube baseline charges the trajectory envelope $\delta + \tilde{\epsilon}_1 \bar{R} \ge \sup_k \psi(\hat{x}_k)$; Theorem 1 charges the invariant-measure average $\bar{\epsilon}_\mu$ plus transients. Theorem 1 and Corollary 4 are not ordered. In the equilibrium case ($\mu = \delta_{x_\star}$, so $\bar{\epsilon}_\mu = \delta$) and the long-horizon limit $\gamma \to 1$, Theorem 1 is the tighter of the two if and only if

$$L_\psi \ \le \ (1 - \bar{\rho}) \frac{\tilde{\epsilon}_1 \bar{R}}{\delta + \tilde{\epsilon}_1 \bar{R}}, \tag{14}$$

in which case its improvement factor over Equation 13 is $(\delta + \tilde{\epsilon}_1 \bar{R})\big/\big(\delta + L_\psi(\delta + \tilde{\epsilon}_1 \bar{R})/(1 - \bar{\rho})\big)$, approaching $(1 - \bar{\rho})/L_\psi$ as $\delta \to 0$ and $1 + \tilde{\epsilon}_1 \bar{R}/\delta$ as $L_\psi \to 0$; above the threshold, the tube baseline is the tighter bound and the transient terms of Equation 3 are redundant. By Remark 1, the regime Equation 14 is the Jacobian-accurate regime, and it is the complement of condition Equation 10 of Corollary 2: the tube baseline governs short horizons and coarse models, while the invariant-measure bound governs the long-horizon, accurate-model regime that motivates this work. The structural difference is where the far-field error is charged: Equation 13 charges the worst error the rollout ever meets at the steady-state rate $1/(1 - \gamma)$, whereas Equation 3 charges the steady-state error $\bar{\epsilon}_\mu$ at that rate and confines the far field to a mixing term that is $O(1)$ in the horizon and a drift term that is second order in the model error.

## 5 Experiments

We report four numerical tests of Theorem 1 (Experiments 1–4). The first three isolate the three structural mechanisms of the bound, measure localization ($\bar{\epsilon}_\mu$ versus $\epsilon_0$), linearised contraction rate ($\bar{\rho}$ versus $\rho$), and finite-sample scaling, on systems with closed-form ground truth, while the fourth exhibits a regime in which the classical bound is infinite and ours remains finite and tight (Figures 1 and 2). Two additional tests appear in the appendix: a neural-network world model on a stabilized pendulum (Section F.5) and a Van der Pol Poincaré-map test of the transverse-contraction extension (Section G.4).

**Experiment 1: measure localization (Panel (a)).** 2D stable linear system $x_{t+1} = Ax_t + \sigma w_t$ with $A = \mathrm{diag}(0.8, 0.7)$, $\sigma = 0.05$. BULK and TAIL perturbation families have matched $\epsilon_0 = c$ but $\bar{\epsilon}_\mu^{\mathrm{TAIL}}/\bar{\epsilon}_\mu^{\mathrm{BULK}} \approx 10^{-2}$. Both collapse on the $\bar{\epsilon}_\mu$ axis and separate by $\sim 20\times$ on the $\epsilon_0$ axis. This isolates the measure-localization mechanism: the system is linear ($\bar{\rho} = \rho$) and the perturbation is $C^\infty$. Full setup and sample-size table in Section F.1.

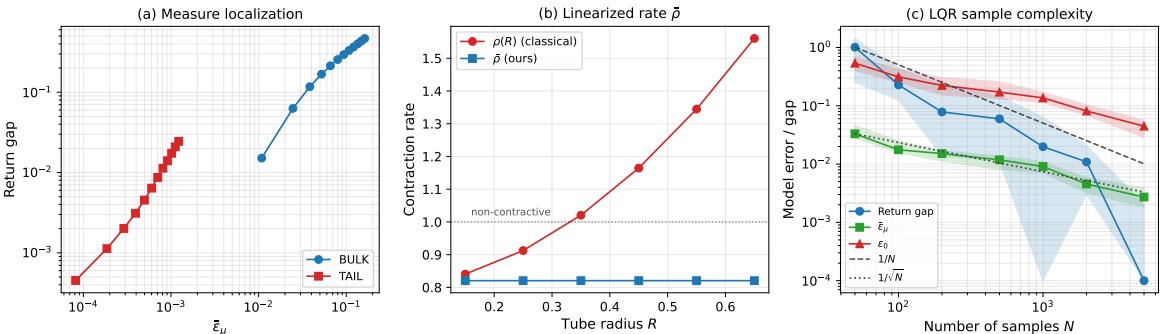

Figure 1: Experiments 1–3 (each panel in log scale). (a) *Experiment 1, measure localization*: BULK and TAIL perturbations with matched $\epsilon_0$ but $100\times$ different $\bar{\epsilon}_\mu$ collapse onto a single line on the $\bar{\epsilon}_\mu$ axis. Shaded bands are $\pm 1$ standard error of the return-gap estimate over 12 Monte Carlo batches; $\epsilon_0$ and $\bar{\epsilon}_\mu$ are computed analytically on a grid and carry no sampling error. (b) *Experiment 2, linearized rate*: $\rho(R)$ crosses unity at $R \approx 0.35$ (classical bound diverges), while $\bar{\rho} \equiv 0.82$ along trajectories. These are deterministic geometric quantities (grid supremum and trajectory-maximum Jacobian norm) and carry no error bar. (c) *Experiment 3, LQR sample complexity*: gap $\propto N^{-1.14}$, $\bar{\epsilon}_\mu \propto N^{-0.499}$; the classical bound has constant $\sim 200$–$300\times$ larger uniformly. Median over 8–12 seeds per $N$; shaded bands are the interquartile range, which widens for the return gap at large $N$ as the gap approaches the Monte Carlo noise floor.

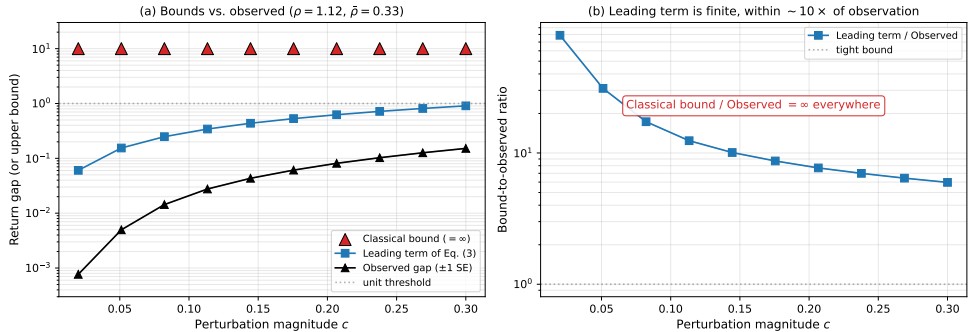

Figure 2: **Experiment 4 (bounds-disagree design).** (a) At tube radius $R = 0.5$, $\rho(R) = 1.12 > 1$, making the classical bound *infinite* at every $c$ (red up-arrows). Our bound (blue, with $\bar{\rho} = 0.33$) is finite and tracks observation (black). The plotted bound curve is the leading term $\gamma L_r \bar{\epsilon}_\mu / [(1-\gamma)(1-\bar{\rho})]$ of Equation 3; the transient magnitudes for this construction are reported in Section F.4. The shaded band on the observed gap is $\pm 1$ standard error over 12 Monte Carlo batches; the analytical bound curves are deterministic and carry no error bar. (b) Our bound is 6–60$\times$ looser than observation; the classical bound's looseness is infinite.

**Experiment 2: linearized contraction rate (Panel (b)).** The cubic map $F(x) = Ax + \alpha \|x\|^2 x$ with $\alpha = 0.6$ has $\rho(R)$ growing quadratically in $R$. A fixed perturbation at the origin yields an observed return gap of $1.3 \times 10^{-2}$, independent of $R$. As $R$ grows past 0.35, $\rho(R) \geq 1$ and the classical bound diverges; our bound, computed with trajectory-measured $\bar{\rho} = 0.82$, remains at $\approx 1.4$, finite and consistent with observation. Full setup in Section F.2.

**Experiment 3: LQR sample complexity (Panel (c)).** Certainty-equivalent LQR from $N$ least-squares samples with 12 seeds per $N$. Fitted log-log slopes over four decades of $N$: return gap $\propto N^{-1.14}$ (theory: $-1$), $\bar{\epsilon}_\mu \propto N^{-0.499}$ (theory: $-0.5$). The classical bound using $\epsilon_0$ has a constant larger than ours by $(\epsilon_0/\bar{\epsilon}_\mu)^2 \approx 200$–300 uniformly. This is the setting of Mania et al. (2019); our bound complements theirs by identifying $\bar{\epsilon}_\mu$ as the explanatory quantity. Full setup in Section F.3.

**Experiment 4: bounds-disagree design.** We construct a system on which the classical bound is *quantitatively vacuous* while ours is finite and predictive. The dynamics are $F(x) = Ax + \alpha \|x\|^2 x$ with $A = \mathrm{diag}(0.30, 0.25)$, $\alpha = 1.1$, $\sigma = 0.02$, $\gamma = 0.95$, over a tube of radius $R = 0.5$. Here $\rho(R) = 1.12$ makes the *classical sup-norm bound infinite*, while trajectory-measured $\bar{\rho} = 0.33$ is strictly below 1. Placing a Gaussian perturbation bump at $3.5\sigma_\mu$ off the origin and sweeping $c \in [0.02, 0.30]$, our bound, plotted as its leading term $\gamma L_r \bar{\epsilon}_\mu / [(1 - \gamma)(1 - \bar{\rho})]$ (see Section F.4 for the transient terms of this construction), ranges from 0.06 to 0.91 and the observed gap from $8 \times 10^{-4}$ to 0.15; our bound is 6–60× looser than observation while the classical bound is infinitely loose. The improvements of Theorem 1 here are qualitative, not incremental. Full setup in Section F.4.

## 6 Limitations and extensions

The bound rests on three assumptions whose scope deserves comment. First, we treat *fixed-policy* evaluation: extending the bound to policy optimization, where the policy is chosen through the learned model, requires controlling the policy-shift term $\left\| \pi_F^\star - \pi_{\hat{F}}^\star \right\|$, and this in turn depends on policy-class-specific sensitivity analyses (closed in the linear- quadratic case via Riccati perturbation; open for general nonlinear classes). Second, the contraction assumption describes a converged stabilizing policy. In the training regime it can be restored by policy-class restriction (Chen et al., 2024; Lawrence et al., 2024), warm-starting from a stabilizing controller (Berkenkamp et al., 2017), or Lyapunov-based shaping of the reward (Westenbroek et al., 2022). Third, although our exposition uses the Euclidean metric, the bound generalizes to any contraction metric (Proposition 1); for systems whose closed loops are not Euclidean- contractive, an appropriate Riemannian metric must be supplied. Contact-rich dynamics violate the local $C^1$ regularity (Pang et al., 2023) and are not covered, although our bound applies directly to smoothed contact models (Todorov, 2011; Pfrommer et al., 2021). Finally, although the bound's constants (the contraction rate $\bar{\rho}$ and the mixing constant $C_{\mathrm{mix}}$) remain dimension-dependent in general, the quantity the bound depends on, the measure-averaged error $\bar{\epsilon}_\mu$, is itself dimension-friendly: It is a Monte Carlo expectation estimated at the $N^{-1/2}$ rate independently of state dimension, whereas certifying the sup-norm error $\epsilon_0$ requires a supremum whose sample complexity grows with dimension. Section F.6 confirms this empirically up to $d = 100$.

## 7 Conclusion

We proved that, for a fixed policy whose closed-loop dynamics on the true system are contractive, the discounted return gap between the true system and a learned dynamics model is controlled by the one-step model error *averaged under the invariant measure of the true closed loop*, rather than by the worst-case supremum of the model error over the state space. The improvement over the classical Lipschitz-MBRL bound of Asadi et al. (2018) comes from three structurally distinct relaxations — measure localization, linearized contraction, and tube-localized smoothness — that are simultaneously active in regimes where the classical bound is vacuous (Figure 2). A direct operational consequence is that the on-policy mean-squared error minimized by modern world-model training admits a return-gap interpretation under closed-loop contraction (Corollary 1). The result extends to stochastic dynamics via a Wasserstein-1 coupling (Section D), and to systems stabilizing a limit cycle via transverse contraction (Section G). The natural next step is the policy-optimization extension, where the policy is itself chosen through the learned model: this introduces a policy-shift term whose treatment depends on the policy class and is left to future work.

### Broader Impact Statement

We prove a return-gap bound for fixed-policy evaluation under a learned dynamics model, which is useful as a diagnostic for sim-to-real verification and for model-based value expansion when closed-loop contraction holds. The principal risk is misapplication: the bound holds only under closed-loop contraction and only for a fixed policy. Practitioners should verify contractivity before relying on the bound's numerical value, and should not extend it to iterative policy optimization without additional sensitivity analysis. We see no dual-use concerns specific to this verification-side result.

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

## A    Appendix

## B    Full proofs for Section 4

This appendix contains the full proofs of Lemmas 1 to 3 and Theorem 1, together with Proposition 1 on the spectral-radius version of the linearized contraction rate.

### B.1    Proof of Lemma 1 (linearized variational recursion)

By the vector-valued fundamental theorem of calculus applied to $F$ on the segment $[\hat{x}_t, x_t] \subseteq \Omega$ (contained in $\Omega$ by convexity and Assumption 4),

$$F(x_t) - F(\hat{x}_t) = \int_0^1 \nabla F(\hat{x}_t + s e_t)\, \mathrm{d}s \cdot e_t = A_t\, e_t. \tag{15}$$

Adding and subtracting $F(\hat{x}_t)$:

$$e_{t+1} = F(x_t) - \hat{F}(\hat{x}_t) = [F(x_t) - F(\hat{x}_t)] + [F(\hat{x}_t) - \hat{F}(\hat{x}_t)] = A_t e_t + \Delta_t. \tag{16}$$

Iterating with $e_0 = 0$ gives $e_t = \sum_{k=0}^{t-1} \Phi_{t,k+1}\, \Delta_k$ where $\Phi_{t,k} := A_{t-1} A_{t-2} \cdots A_k$ and $\Phi_{t,t} := I_d$. The operator-norm bound $\|A_t\| \le \bar{\rho}$ follows from the Bochner-integral triangle inequality $\|A_t\| \le \int_0^1 \|\nabla F(\hat{x}_t + s e_t)\|\, \mathrm{d}s \le \bar{\rho}$, and submultiplicativity gives $\|\Phi_{t,k}\| \le \bar{\rho}^{\,t-k}$. Substituting yields $\|e_t\| \le \sum_{k=0}^{t-1} \bar{\rho}^{\,t-1-k}\, \|\Delta_k\|$. $\qquad\square$

**Proposition 1** (Spectral Jacobian product bound with contraction metric)**.** *Suppose there exists a symmetric positive-definite $M \in \mathbb{R}^{d \times d}$ with condition number $\kappa := \lambda_{\max}(M)/\lambda_{\min}(M)$ and $\rho_M \in [0, 1)$ such that $A^\top M A \preceq \rho_M^2 M$ for all $A \in \text{conv}\{\nabla F(\xi) : \xi \in \Omega\}$. Then*

$$\left\| \Phi_{t,k} \right\|_{\text{op}} \ \leq \ \sqrt{\kappa}\, \rho_M^{t-k}. \tag{17}$$

*Proof.* Each $A_t$ is a Bochner integral of Jacobian values and therefore lies in the convex hull $\text{conv}\{\nabla F(\xi) : \xi \in \Omega\}$, so $A_t^\top M A_t \preceq \rho_M^2 M$. In the $M$-norm $\|v\|_M := \sqrt{v^\top M v}$, this gives $\|A_t v\|_M \leq \rho_M \|v\|_M$ and hence $\|\Phi_{t,k} v\|_M \leq \rho_M^{t-k} \|v\|_M$. By norm equivalence, $\|v\|_M^2 \in [\lambda_{\min}(M), \lambda_{\max}(M)] \cdot \|v\|^2$, so $\|\Phi_{t,k} v\|^2 \leq \lambda_{\min}(M)^{-1} \|\Phi_{t,k} v\|_M^2 \leq \lambda_{\min}(M)^{-1} \rho_M^{2(t-k)} \lambda_{\max}(M) \|v\|^2 = \kappa \rho_M^{2(t-k)} \|v\|^2$. $\qquad \square$

[Proposition 1](#) lets us replace the Euclidean contraction rate $\bar{\rho}$ in [Theorem 1](#) by the $M$-contraction rate $\rho_M$, at the cost of a constant $\sqrt{\kappa}$ in front. This is useful for non-normal systems (such as the linearized pendulum around upright, which has $\|A\|_2 > 1$ but $\rho(A) < 1$) where the Euclidean rate is vacuous but a weighted-norm rate is not.

## B.2 Proof of [Lemma 2](#) (trajectory-tube one-step error)

Apply the vector-valued fundamental theorem of calculus to $F - \hat{F}$ on the segment $[x_\star, \hat{x}_k] \subseteq \mathcal{T}$:

$$(F - \hat{F})(\hat{x}_k) - (F - \hat{F})(x_\star) = \int_0^1 [\nabla F - \nabla \hat{F}](x_\star + s(\hat{x}_k - x_\star))\, \mathrm{d}s \cdot (\hat{x}_k - x_\star). \tag{18}$$

Taking norms,

$$\|\Delta_k\| = \left\| (F - \hat{F})(\hat{x}_k) \right\| \leq \left\| (F - \hat{F})(x_\star) \right\| + \sup_{\xi \in \mathcal{T}} \left\| \nabla F(\xi) - \nabla \hat{F}(\xi) \right\| \cdot \|\hat{x}_k - x_\star\| = \delta + \tilde{\epsilon}_1 \|\hat{x}_k - x_\star\|. \quad \square \tag{19}$$

## B.3 Proof of [Lemma 3](#) (measure transfer)

For deterministic $F$, the pushforward $\delta_{x_0} F^t$ of the point mass $\delta_{x_0}$ under the $t$-fold composition of $F$ is the point mass $\delta_{x_t}$ at the trajectory state $x_t$. By Kantorovich–Rubinstein duality, for any $L_\psi$-Lipschitz test function $\psi$,

$$|\psi(x_t) - \mathbb{E}_\mu \psi| = |*| \int \psi\, \mathrm{d}\delta_{x_t} - \int \psi\, \mathrm{d}\mu \leq L_\psi\, W_1(\delta_{x_t}, \mu) \leq L_\psi\, C_{\text{mix}}\, \alpha^t, \tag{20}$$

using assumption (a) of [Theorem 1](#) in the last inequality. $\qquad \square$

## B.4 Proof of [Theorem 1](#)

From [Lemma 1](#), $\|e_t\| \leq \sum_{k=0}^{t-1} \bar{\rho}^{t-1-k} \|\Delta_k\|$. Combining with the Lipschitz-reward [Assumption 3](#) and $e_0 = 0$,

$$\left| J(F) - J(\hat{F}) \right| \leq L_r \sum_{t=1}^{\infty} \gamma^t \|e_t\| \leq L_r \sum_{t=1}^{\infty} \gamma^t \sum_{k=0}^{t-1} \bar{\rho}^{t-1-k} \|\Delta_k\|. \tag{21}$$

Swap summation order (Tonelli applies since all terms are non-negative) with index change $s = t - 1 - k$:

$$\left| J(F) - J(\hat{F}) \right| \leq L_r \sum_{k=0}^{\infty} \|\Delta_k\| \sum_{t=k+1}^{\infty} \gamma^t \bar{\rho}^{t-1-k} = \frac{\gamma L_r}{1 - \gamma \bar{\rho}} \sum_{k=0}^{\infty} \gamma^k \|\Delta_k\|. \tag{22}$$

It remains to bound $\sum_{k=0}^{\infty} \gamma^k \|\Delta_k\|$. Writing $\|\Delta_k\| = \psi(\hat{x}_k)$, decompose

$$\psi(\hat{x}_k) = \underbrace{[\psi(\hat{x}_k) - \psi(x_k)]}_{\text{(I)}} + \underbrace{[\psi(x_k) - \mathbb{E}_\mu \psi]}_{\text{(II)}} + \underbrace{\mathbb{E}_\mu \psi}_{\text{(III)}=\bar{\epsilon}_\mu}. \tag{23}$$

*Bounding (I)*. By $L_\psi$-Lipschitz continuity of $\psi$ on $\Omega$, $|\psi(\hat{x}_k) - \psi(x_k)| \leq L_\psi \|\hat{x}_k - x_k\|$. Using Lemma 1 and Lemma 2 with $\bar{R} = \sup_k \|\hat{x}_k - x_\star\|$,

$$\|\hat{x}_k - x_k\| \leq \sum_{j=0}^{k-1} \bar{\rho}^{k-1-j} \|\Delta_j\| \leq \sum_{j=0}^{k-1} \bar{\rho}^{k-1-j} (\delta + \tilde{\epsilon}_1 \bar{R}) \leq \frac{\delta + \tilde{\epsilon}_1 \bar{R}}{1 - \bar{\rho}}, \tag{24}$$

so $|\psi(\hat{x}_k) - \psi(x_k)| \leq L_\psi(\delta + \tilde{\epsilon}_1 \bar{R})/(1 - \bar{\rho})$.

*Bounding (II)*. By Lemma 3, $|\psi(x_k) - \mathbb{E}_\mu \psi| \leq L_\psi C_{\mathrm{mix}} \alpha^k$.

*Bounding (III)*. Equals $\bar{\epsilon}_\mu$ by definition.

Substituting (I), (II), (III) back into Equation 22:

$$\left| J(F) - J(\hat{F}) \right| \leq \frac{\gamma L_r}{1 - \gamma \bar{\rho}} \sum_{k=0}^{\infty} \gamma^k \left[ \bar{\epsilon}_\mu + \frac{L_\psi(\delta + \tilde{\epsilon}_1 \bar{R})}{1 - \bar{\rho}} + L_\psi C_{\mathrm{mix}} \alpha^k \right]$$

$$= \frac{\gamma L_r}{1 - \gamma \bar{\rho}} \left[ \frac{\bar{\epsilon}_\mu}{1 - \gamma} + \frac{L_\psi(\delta + \tilde{\epsilon}_1 \bar{R})}{(1 - \bar{\rho})(1 - \gamma)} + \frac{L_\psi C_{\mathrm{mix}}}{1 - \gamma \alpha} \right]. \tag{25}$$

Since $\gamma \in (0, 1)$ implies $(1 - \gamma \bar{\rho}) \geq (1 - \bar{\rho})$, we have $1/(1 - \gamma \bar{\rho}) \leq 1/(1 - \bar{\rho})$. Upper-bounding the prefactor in this way and factoring out $\gamma L_r / [(1 - \gamma)(1 - \bar{\rho})]$ yields

$$\left| J(F) - J(\hat{F}) \right| \leq \frac{\gamma L_r}{(1 - \gamma)(1 - \bar{\rho})} \left( \bar{\epsilon}_\mu + \frac{L_\psi(\delta + \tilde{\epsilon}_1 \bar{R})}{1 - \bar{\rho}} + \frac{L_\psi C_{\mathrm{mix}}(1 - \gamma)}{1 - \gamma \alpha} \right), \tag{26}$$

which is Equation 3. $\qquad\square$

### B.5  Tightness of $\bar{\rho}$ versus $\rho$

Lemma 1 uses $\bar{\rho} = \sup_t \sup_{\xi \in [\hat{x}_t, x_t]} \|\nabla F(\xi)\|$, which is always at most the global Lipschitz rate $\rho$ from Assumption 2. The inequality is strict when $\nabla F$ varies and the trajectory-error segments $[\hat{x}_t, x_t]$ avoid the regions of $\Omega$ where $\|\nabla F\|$ attains its maximum. For a nonlinear system stabilized near an equilibrium, $\|\nabla F\|$ typically peaks far from the equilibrium (where nonlinearities are strongest), while the trajectories concentrate near the equilibrium. The ratio $\rho/\bar{\rho}$ can therefore be order-wise larger in nonlinear regimes and quantifies the "nonlinear slack" the problem admits.

## C  Concentration bound for Corollary 1

We spell out the concentration step underlying Corollary 1. The empirical loss is the squared error $\left\| F(x_i) - \hat{F}_\theta(x_i) \right\|^2$, so the relevant function class is

$$\mathcal{G} = \left\{ x \mapsto \left\| F(x) - \hat{F}_\theta(x) \right\|^2 : \theta \in \Theta \right\}. \tag{27}$$

Each element of $\mathcal{G}$ is bounded by $B^2$ where $B := \sup_{x \in \Omega, \theta \in \Theta} \left\| F(x) - \hat{F}_\theta(x) \right\|$. When $\theta \mapsto \hat{F}_\theta$ is $L_\theta$-Lipschitz in parameters and $\left\| F - \hat{F}_\theta \right\|$ is itself $L_\psi$-Lipschitz in $x$, the squared-error class $\mathcal{G}$ has Lipschitz constant at most $2BL_\psi$ in $x$ (by chain rule on $t \mapsto t^2$ for $t \leq B$). Let $\mathfrak{R}_N(\mathcal{G})$ denote the Rademacher complexity of $\mathcal{G}$ on $N$ i.i.d. samples from $\mu$.

**Proposition 2** (Concentration of the empirical squared error). *For any $\eta \in (0, 1)$, with probability at least $1 - \eta$ over $N$ i.i.d. samples $x_1, \ldots, x_N \sim \mu$, uniformly over $\theta \in \Theta$,*

$$\mathbb{E}_{x \sim \mu} \left\| F(x) - \hat{F}_\theta(x) \right\|^2 \leq \frac{1}{N} \sum_{i=1}^{N} \left\| F(x_i) - \hat{F}_\theta(x_i) \right\|^2 + 2\mathfrak{R}_N(\mathcal{G}) + B^2 \sqrt{\frac{\log(2/\eta)}{2N}}. \tag{28}$$

*Proof sketch.* Standard symmetrization + McDiarmid applied to the bounded class $\mathcal{G}$; see, e.g., Chapter 3 of Mohri et al. (2018) or Chapter 4 of Wainwright (2019). The bounded-difference constant is $B^2/N$. $\square$

For an $L_\psi$-Lipschitz original class on compact $\Omega \subset \mathbb{R}^d$ of diameter $D$, the squared-error class has Lipschitz constant $\leq 2BL_\psi$, so $\mathfrak{R}_N(\mathcal{G}) = O(BL_\psi D/\sqrt{N})$ up to logarithmic factors by Dudley's entropy integral (Wainwright, 2019).

Combining with Jensen's inequality

$$\bar{\epsilon}_\mu = \mathbb{E}_\mu \left\| F - \hat{F}_\theta \right\| \leq \sqrt{\mathbb{E}_\mu \left\| F - \hat{F}_\theta \right\|^2} \tag{29}$$

and substituting Equation 28 gives Corollary 1's bound:

$$\bar{\epsilon}_\mu \leq \sqrt{\hat{\mathcal{L}}_N + 2\mathfrak{R}_N(\mathcal{G}) + B^2\sqrt{\log(2/\eta)/(2N)}}. \tag{30}$$

The convergence rate is $\bar{\epsilon}_\mu - \sqrt{\hat{\mathcal{L}}_N} = O(N^{-1/4})$, slower than the $O(N^{-1/2})$ rate of the unsquared class because the square-root composition halves the effective concentration rate. This is the unavoidable cost of working with the empirical MSE rather than empirical mean absolute error.

## D  Stochastic extension of Theorem 1

We extend Theorem 1 to stochastic closed-loop dynamics. The main theorem assumed deterministic maps $F, \hat{F} : \Omega \to \Omega$. We now replace them with stochastic transition kernels $P, \hat{P}$ on $\Omega$, and show that the return-gap bound survives with the same structural form. The key technical subtlety is that the error $e_t = x_t - \hat{x}_t$ is now a random variable, so the deterministic variational recursion of Lemma 1 becomes an equation on random variables driven by two coupled noise processes. We handle this via a synchronous coupling.

### D.1  Setup

Let $P(\cdot \mid x)$ and $\hat{P}(\cdot \mid x)$ be Borel transition kernels on $\Omega \subseteq \mathbb{R}^d$. Trajectories are generated by $x_{t+1} \sim P(\cdot \mid x_t)$ and $\hat{x}_{t+1} \sim \hat{P}(\cdot \mid \hat{x}_t)$ with $x_0 = \hat{x}_0 \in \Omega$. The discounted return is

$$J(P) := \mathbb{E}\Big[\sum_{t=0}^{\infty} \gamma^t r(x_t) \,\Big|\, x_0\Big], \qquad J(\hat{P}) := \mathbb{E}\Big[\sum_{t=0}^{\infty} \gamma^t r(\hat{x}_t) \,\Big|\, \hat{x}_0\Big], \tag{31}$$

where the expectations are over the respective kernels.

### D.2  Assumptions

We replace Assumption 2 (deterministic contraction) and Assumption 5 (deterministic $C^1$ error) by stochastic analogs.

**Assumption 6** (Wasserstein-1 contraction of $P$)**.** There exists $\rho \in [0, 1)$ such that for all $x, y \in \Omega$,

$$W_1\big(P(\cdot \mid x), P(\cdot \mid y)\big) \leq \rho \left\| x - y \right\|. \tag{32}$$

**Assumption 7** (Kernel-level model error)**.** The one-step kernel discrepancy $\psi(x) := W_1\big(P(\cdot \mid x), \hat{P}(\cdot \mid x)\big)$ is $L_\psi$-Lipschitz on $\Omega$.

**Assumption 8** (Ergodicity of $P$)**.** $P$ admits a unique invariant probability measure $\mu$ satisfying $W_1(\delta_x P^t, \mu) \leq C_{\text{mix}}\alpha^t$ for some $\alpha \in [0, 1)$ and $C_{\text{mix}} \leq \text{diam}(\Omega)$.

We retain Assumptions 1, 3 and 4 from the deterministic setting. The $\mu$-averaged model error is

$$\bar{\epsilon}_\mu := \mathbb{E}_{x \sim \mu}\big[\psi(x)\big] = \mathbb{E}_{x \sim \mu}\big[W_1(P(\cdot \mid x), \hat{P}(\cdot \mid x))\big]. \tag{33}$$

### D.3 Synchronous coupling

The deterministic proof of Theorem 1 used the identity $e_{t+1} = F(x_t) - \hat{F}(\hat{x}_t)$ pointwise. Stochastically, $x_{t+1}$ and $\hat{x}_{t+1}$ are random, and the joint law is not determined by the marginal kernels alone, it depends on how we *couple* the two processes. We use a synchronous coupling that realizes the Wasserstein distance at each step.

**Lemma 4** (Synchronous Wasserstein-optimal coupling). *Under Assumptions 6 and 7, there exists a joint distribution $\Pi$ on $(\Omega \times \Omega)^{\mathbb{N}}$ such that the marginal of the first component is a $P$-trajectory from $x_0$, the marginal of the second is a $\hat{P}$-trajectory from $\hat{x}_0$, and for every $t \geq 0$,*

$$\mathbb{E}_{\Pi}\big[\|x_{t+1} - \hat{x}_{t+1}\| \,\big|\, x_t, \hat{x}_t\big] \;\leq\; \rho\,\|x_t - \hat{x}_t\| + \psi(\hat{x}_t). \tag{34}$$

*Proof.* At each step $t$, given $(x_t, \hat{x}_t)$, we construct the coupling of $P(\cdot \mid x_t)$ and $\hat{P}(\cdot \mid \hat{x}_t)$ as follows. First, the Wasserstein-optimal coupling $\Pi_1$ of $P(\cdot \mid x_t)$ and $P(\cdot \mid \hat{x}_t)$ satisfies $\mathbb{E}_{\Pi_1}\|x_{t+1} - y\| = W_1(P(\cdot \mid x_t), P(\cdot \mid \hat{x}_t)) \leq \rho\|x_t - \hat{x}_t\|$ by Assumption 6. Second, the Wasserstein-optimal coupling $\Pi_2$ of $P(\cdot \mid \hat{x}_t)$ and $\hat{P}(\cdot \mid \hat{x}_t)$ satisfies $\mathbb{E}_{\Pi_2}\|y - \hat{x}_{t+1}\| = W_1(P(\cdot \mid \hat{x}_t), \hat{P}(\cdot \mid \hat{x}_t)) = \psi(\hat{x}_t)$ by Assumption 7. The gluing lemma for Wasserstein couplings (Villani, 2009, Lemma 7.6) gives a joint distribution on $(x_{t+1}, y, \hat{x}_{t+1})$ with the correct marginals on pairs. Integrating out $y$ and applying the triangle inequality,

$$\mathbb{E}\,\|x_{t+1} - \hat{x}_{t+1}\| \leq \mathbb{E}\,\|x_{t+1} - y\| + \mathbb{E}\,\|y - \hat{x}_{t+1}\| \leq \rho\,\|x_t - \hat{x}_t\| + \psi(\hat{x}_t).$$

Iterating this construction at each step gives the full-trajectory coupling $\Pi$. $\qquad\square$

### D.4 Stochastic return-gap bound

**Theorem 2** (Stochastic invariant-measure return-gap bound). *Let $\beta := \max(\rho, \alpha)$. Under Assumptions 1, 3, 4 and 6 to 8 and the geometric condition $\gamma L_\psi < 1 - \gamma\beta$, for any $\gamma \in (0,1)$ and any shared initial state $x_0 = \hat{x}_0 \in \Omega$,*

$$\big|J(P) - J(\hat{P})\big| \;\leq\; \frac{\gamma L_r}{1 - \gamma\rho} \cdot \frac{1}{1 - \dfrac{\gamma L_\psi}{1 - \gamma\beta}} \left[\frac{\bar{\epsilon}_\mu}{1 - \gamma} + \frac{L_\psi\,C_{\mathrm{mix}}}{1 - \gamma\beta}\right]. \tag{35}$$

The sharp condition $\gamma L_\psi < 1 - \gamma\beta$ (equivalently $L_\psi < (1 - \gamma\beta)/\gamma$) is exactly what the proof requires; it is implied by the simpler sufficient condition $L_\psi < 1 - \beta$, which we use to state a cleaner corollary below.

**Corollary 5** (Simplified $\beta$-bound). *Under the additional simplifying assumption $L_\psi < 1 - \beta$, the bound Equation 35 simplifies to*

$$\big|J(P) - J(\hat{P})\big| \;\leq\; \frac{\gamma L_r}{(1 - \gamma)(1 - \rho)(1 - L_\psi/(1 - \beta))} \left(\bar{\epsilon}_\mu + \frac{L_\psi C_{\mathrm{mix}}(1 - \gamma)}{1 - \gamma\beta}\right). \tag{36}$$

*This is the form most directly comparable to Theorem 1: the dominant term is $\bar{\epsilon}_\mu$ with prefactor $\gamma L_r/[(1 - \gamma)(1 - \rho)(1 - L_\psi/(1 - \beta))]$, the mixing transient decays at rate $\beta$, and $\beta = \rho$ when contraction is slower than mixing (recovering the deterministic-style prefactor exactly).*

*Proof.* Let $\Pi$ be the synchronous coupling from Lemma 4, and write $\mathbb{E}$ for expectation under $\Pi$. Let $e_t := \mathbb{E}\,\|x_t - \hat{x}_t\|$ be the expected trajectory error. From Equation 34 and iterated expectation,

$$e_{t+1} \;=\; \mathbb{E}\,\|x_{t+1} - \hat{x}_{t+1}\| \;\leq\; \rho\,\mathbb{E}\,\|x_t - \hat{x}_t\| + \mathbb{E}\psi(\hat{x}_t) \;=\; \rho\,e_t + \mathbb{E}\psi(\hat{x}_t). \tag{37}$$

With $e_0 = 0$, unrolling gives

$$e_t \;\leq\; \sum_{k=0}^{t-1} \rho^{t-1-k}\,\mathbb{E}\psi(\hat{x}_k). \tag{38}$$

The return gap bound is

$$
\begin{aligned}
\left| J(P) - J(\hat{P}) \right| = |*| \mathbb{E} \sum_{t=0}^{\infty} \gamma^t \big( r(x_t) - r(\hat{x}_t) \big) \\
\leq L_r \sum_{t=0}^{\infty} \gamma^t e_t \\
\leq L_r \sum_{t=1}^{\infty} \gamma^t \sum_{k=0}^{t-1} \rho^{t-1-k} \, \mathbb{E}\psi(\hat{x}_k) \\
= \frac{\gamma L_r}{1 - \gamma\rho} \sum_{k=0}^{\infty} \gamma^k \, \mathbb{E}\psi(\hat{x}_k),
\end{aligned}
\tag{39}
$$

where the last step uses Tonelli (non-negative terms) and the same index shift as in the deterministic proof. It remains to bound $\mathbb{E}\psi(\hat{x}_k)$. Let $x_k$ be the $P$-trajectory from $x_0$ under the coupling. Decompose

$$
\mathbb{E}\psi(\hat{x}_k) \;=\; \mathbb{E}[\psi(\hat{x}_k) - \psi(x_k)] + \mathbb{E}[\psi(x_k) - \mathbb{E}_\mu \psi] + \mathbb{E}_\mu \psi.
\tag{40}
$$

*Term II (mixing transient).* Since $x_k$ is a $P$-trajectory from $x_0$ and $\psi$ is $L_\psi$-Lipschitz, Kantorovich–Rubinstein duality gives

$$
\left| \mathbb{E}\psi(x_k) - \mathbb{E}_\mu \psi \right| \;\leq\; L_\psi \, W_1\big(\mathrm{Law}(x_k), \mu\big) \;\leq\; L_\psi \, C_{\mathrm{mix}} \, \alpha^k.
\tag{41}
$$

*Term III.* Equals $\bar{\epsilon}_\mu$ by definition.

*Term I (model-drift).* This is subtler than the deterministic case because the drift bound Equation 38 itself involves $\mathbb{E}\psi(\hat{x}_j)$ for $j < k$, creating a self-referential inequality. We resolve it as follows. Let $u_k := \mathbb{E}\psi(\hat{x}_k)$. Combining Equation 40 with Equation 41, Term III, and $L_\psi$-Lipschitz continuity applied to Equation 38,

$$
u_k \;\leq\; \bar{\epsilon}_\mu + L_\psi C_{\mathrm{mix}} \alpha^k + L_\psi \sum_{j=0}^{k-1} \rho^{k-1-j} \, u_j.
\tag{42}
$$

This is a discrete-time convolution inequality. Rather than attempting a pointwise bound on $u_k$, which fails when the contraction rate $\rho$ and mixing rate $\alpha$ differ, we compute the discounted sum directly. Define $\beta := \max(\rho, \alpha)$. From Equation 42 and $\rho^{k-1-j} \leq \beta^{k-1-j}$,

$$
u_k \;\leq\; v_k + L_\psi \sum_{j=0}^{k-1} \beta^{k-1-j} \, u_j, \qquad \text{where} \quad v_k := \bar{\epsilon}_\mu + L_\psi C_{\mathrm{mix}} \beta^k.
\tag{43}
$$

Multiply both sides by $\gamma^k$ and sum over $k \geq 0$:

$$
\begin{aligned}
S := \sum_{k=0}^{\infty} \gamma^k u_k \leq \sum_{k=0}^{\infty} \gamma^k v_k + L_\psi \sum_{k=0}^{\infty} \gamma^k \sum_{j=0}^{k-1} \beta^{k-1-j} \, u_j \\
= V + L_\psi \sum_{j=0}^{\infty} u_j \sum_{k=j+1}^{\infty} \gamma^k \beta^{k-1-j} \quad \text{(Fubini)} \\
= V + L_\psi \sum_{j=0}^{\infty} u_j \cdot \frac{\gamma^{j+1}}{1 - \gamma\beta} \quad \text{(geometric sum in } k) \\
= V + \frac{\gamma L_\psi}{1 - \gamma\beta} \, S,
\end{aligned}
\tag{44}
$$

where $V := \sum_{k \geq 0} \gamma^k v_k = \bar{\epsilon}_\mu/(1-\gamma) + L_\psi C_{\mathrm{mix}}/(1-\gamma\beta)$. Provided the geometric condition $\gamma L_\psi < 1 - \gamma\beta$ holds (which is implied by the stronger $L_\psi < 1 - \beta$ since $\gamma < 1$), Equation 44 can be solved for $S$:

$$S \;\leq\; \frac{V}{1 - \gamma L_\psi/(1-\gamma\beta)} \;=\; \frac{(1-\gamma\beta)\,V}{1-\gamma\beta - \gamma L_\psi}. \tag{45}$$

Substituting $S$ into Equation 39 gives directly

$$\left|J(P) - J(\hat{P})\right| \;\leq\; \frac{\gamma L_r}{1-\gamma\rho} \cdot S \;\leq\; \frac{\gamma L_r}{1-\gamma\rho} \cdot \frac{V}{1-\gamma L_\psi/(1-\gamma\beta)}, \tag{46}$$

which is the sharp form Equation 35. Substituting the simpler sufficient condition $L_\psi < 1 - \beta$ gives $1 - \gamma L_\psi/(1-\gamma\beta) \geq (1-\gamma)(1-L_\psi/(1-\beta))$ (algebra), yielding the simplified form Equation 36 of Corollary 5. $\quad\square$

### D.5  Comparison to the deterministic case

Theorem 2 recovers the structural form of Theorem 1 with three changes:

(i) The contraction rate $\rho$ in the stochastic case is the Wasserstein-1 contraction rate of the kernel, not a linearized Jacobian rate. In the deterministic limit where $P(\cdot \mid x) = \delta_{F(x)}$, Assumption 6 reduces to $\|F(x) - F(y)\| \leq \rho \|x - y\|$ and Theorem 2 reduces to the $C^0$ version of Theorem 1 (without the averaged-Jacobian sharpening).

(ii) The model error $\psi(x) = W_1(P(\cdot \mid x), \hat{P}(\cdot \mid x))$ is a kernel-level Wasserstein distance, which reduces to the pointwise norm $\left\|F(x) - \hat{F}(x)\right\|$ in the deterministic limit.

(iii) The anchor and tube-$C^1$ sharpenings of the deterministic case do not appear directly; they can be recovered by an additional assumption that $\psi$ admits an anchored-Lipschitz decomposition $\psi(x) \leq \delta + \epsilon_1 \|x - x_\star\|$, which is a stochastic analog of Assumption 5.

The essential structural claim, that the dominant model-error term is $\bar{\epsilon}_\mu$, an expectation under the true invariant measure, survives verbatim. This is the sense in which the deterministic result is not specific to deterministic dynamics.

## E   Wasserstein distance between the true and learned invariant measures

Theorem 1 bounds the return gap using $\bar{\epsilon}_\mu = \mathbb{E}_\mu \left\|F - \hat{F}\right\|$, the expected model error under the *true* closed-loop invariant measure $\mu$. An agent planning inside the learned model $\hat{F}$, however, simulates trajectories that equilibrate to the learned-model invariant measure $\hat{\mu}$, not to $\mu$. This appendix bounds the Wasserstein distance $W_1(\mu, \hat{\mu})$ in terms of the same quantity $\bar{\epsilon}_\mu$ that controls the return gap, establishing a structural symmetry: small on-policy model error simultaneously controls both the return gap and the induced measure shift.

### E.1  Main result

**Theorem 3** (Invariant-measure Wasserstein bound)**.** *Assume* $F, \hat{F} : \Omega \to \Omega$ *are measurable,* $F$ *is* $\bar{\rho}$-*Wasserstein-contractive in the sense that* $W_1(\delta_x F, \delta_y F) \leq \bar{\rho} \|x - y\|$ *for all* $x, y \in \Omega$, $\hat{F}$ *admits an invariant measure* $\hat{\mu}$ *on* $\Omega$, *and the one-step error* $\psi(x) := \left\|F(x) - \hat{F}(x)\right\|$ *is* $L_\psi$-*Lipschitz on* $\Omega$ *with* $L_\psi < 1 - \bar{\rho}$. *Then*

$$W_1(\mu, \hat{\mu}) \;\leq\; \frac{\bar{\epsilon}_\mu}{1 - \bar{\rho} - L_\psi}, \qquad where \quad \bar{\epsilon}_\mu := \mathbb{E}_{x \sim \mu} \left\|F(x) - \hat{F}(x)\right\|. \tag{47}$$

*In particular, when* $L_\psi = 0$ *(the kernel error has no Lipschitz dependence on state, e.g., constant additive perturbation), the denominator simplifies to* $1 - \bar{\rho}$.

*Proof.* The Wasserstein distance satisfies the contraction-perturbation inequality for pushforward measures. Let $F_{\#}\mu$ and $\hat{F}_{\#}\nu$ denote the pushforwards of measures $\mu, \nu$ under $F, \hat{F}$ respectively. We first establish a one-step perturbation bound.

*Step 1: one-step perturbation.* For any probability measure $\nu$ on $\Omega$,

$$W_1(F_{\#}\nu, \hat{F}_{\#}\nu) \ \leq \ \mathbb{E}_{x\sim\nu} \left\| F(x) - \hat{F}(x) \right\|. \tag{48}$$

This follows from the coupling definition: the map $x \mapsto (F(x), \hat{F}(x))$ pushes $\nu$ to a joint distribution on $\Omega \times \Omega$ with marginals $F_{\#}\nu$ and $\hat{F}_{\#}\nu$, and whose Wasserstein cost $\mathbb{E}_\nu \left\| F(x) - \hat{F}(x) \right\|$ upper bounds $W_1(F_{\#}\nu, \hat{F}_{\#}\nu)$ by definition of the Wasserstein-1 distance.

*Step 2: triangle and contraction.* Since $\mu$ is $F$-invariant and $\hat{\mu}$ is $\hat{F}$-invariant,

$$
\begin{aligned}
W_1(\mu, \hat{\mu}) &= W_1(F_{\#}\mu, \hat{F}_{\#}\hat{\mu}) \\
&\leq W_1(F_{\#}\mu, F_{\#}\hat{\mu}) + W_1(F_{\#}\hat{\mu}, \hat{F}_{\#}\hat{\mu}) \\
&\leq \bar{\rho}\, W_1(\mu, \hat{\mu}) + \mathbb{E}_{x\sim\hat{\mu}} \left\| F(x) - \hat{F}(x) \right\|,
\end{aligned}
\tag{49}
$$

where the second line uses the triangle inequality for Wasserstein distance, and the third line uses the Wasserstein contraction of $F$ on the first term and Equation 48 with $\nu = \hat{\mu}$ on the second.

*Step 3: close the recursion.* Let $\bar{\epsilon}_{\hat{\mu}} := \mathbb{E}_{x\sim\hat{\mu}} \left\| F(x) - \hat{F}(x) \right\|$. From Equation 49,

$$(1 - \bar{\rho})\, W_1(\mu, \hat{\mu}) \ \leq \ \bar{\epsilon}_{\hat{\mu}}. \tag{50}$$

*Step 4: transfer to $\bar{\epsilon}_\mu$.* It remains to relate $\bar{\epsilon}_{\hat{\mu}}$ to $\bar{\epsilon}_\mu$. Assume $\psi(x) := \left\| F(x) - \hat{F}(x) \right\|$ is $L_\psi$-Lipschitz on $\Omega$. Then

$$\bar{\epsilon}_{\hat{\mu}} - \bar{\epsilon}_\mu = \int \psi\, d\hat{\mu} - \int \psi\, d\mu \ \leq \ L_\psi \cdot W_1(\mu, \hat{\mu}). \tag{51}$$

Substituting Equation 51 into Equation 50 and rearranging,

$$(1 - \bar{\rho} - L_\psi)\, W_1(\mu, \hat{\mu}) \ \leq \ \bar{\epsilon}_\mu. \tag{52}$$

Under the assumption $L_\psi < 1 - \bar{\rho}$, this gives

$$W_1(\mu, \hat{\mu}) \ \leq \ \frac{\bar{\epsilon}_\mu}{1 - \bar{\rho} - L_\psi}. \tag{53}$$

When $L_\psi = 0$, the denominator simplifies to $1 - \bar{\rho}$. $\qquad\square$

### E.2 Consequences

**Corollary 6** (Equivalence of on-policy and off-policy training objectives). *Under the hypotheses of Theorem 3,*

$$\left| \bar{\epsilon}_\mu - \bar{\epsilon}_{\hat{\mu}} \right| \ \leq \ \frac{L_\psi\, \bar{\epsilon}_\mu}{1 - \bar{\rho} - L_\psi}. \tag{54}$$

*In particular, if training minimizes the on-policy empirical loss $\hat{\mathcal{L}}_N = \frac{1}{N}\sum_{i=1}^{N} \psi(\hat{x}_i)^2$ with $\hat{x}_i \sim \hat{\mu}$ (samples from the learned-model invariant distribution, as obtained by rolling out the learned model forward in time), the quantity minimized concentrates on $\bar{\epsilon}_{\hat{\mu}}$, which differs from $\bar{\epsilon}_\mu$ by at most $O(L_\psi \bar{\epsilon}_\mu)$. Thus minimizing the on-policy loss under the* learned *model is a provably good surrogate for minimizing the loss under the* true *invariant measure, as long as the error function is not too rough.*

*Proof.* Apply Equation 51 and Equation 53. $\qquad\square$

### E.3 Interpretation

Theorem 3 closes a structural gap in the main theorem. Theorem 1 bounds the return gap in terms of the expected model error under the *true* invariant measure $\mu$, but an agent using the learned model for planning cannot sample from $\mu$, it can only sample from $\hat{\mu}$, the invariant measure of the learned model. Theorem 3 shows that $\mu$ and $\hat{\mu}$ are close in Wasserstein-1 distance, with the bound scaling in the same quantity $(\bar{\epsilon}_\mu)$ that controls the return gap. This has two practical consequences:

*Consistency of on-policy training.* The empirical loss minimized by on-policy world-model training is $\mathbb{E}_{\hat{\mu}}\psi^2$, not $\mathbb{E}_\mu\psi^2$. Corollary 6 shows the two are within $O(L_\psi\bar{\epsilon}_\mu)$ of each other, so the training objective remains a valid upper bound on the return-gap-controlling quantity.

*No catastrophic hallucination.* If the return-gap bound is small ($\bar{\epsilon}_\mu$ is small), then the learned invariant measure cannot drift arbitrarily far from the true invariant measure: the same $\bar{\epsilon}_\mu$ that certifies return-gap smallness also certifies measure-shift smallness. This addresses a standard objection to MBRL: that optimizing policies through a learned model can exploit hallucinated dynamics. Our bound shows that when the model is accurate on-policy, the long-horizon attractors of the learned and true dynamics are close in the Wasserstein sense.

## F  Experimental details and additional experiment

This appendix provides reproducibility details for Experiments 1–4 (in Section 5) and presents Experiment 5 in full. Experiment 6 (Van der Pol limit cycle, verifying the transverse-contraction extension) appears in Section G.4. Experiment 7 (high-dimensional tractability) appears in Section F.6 below.

**Overview.** The seven experiments collectively probe Theorem 1 and its two extensions. Table 1 summarises what each tests and the principal result. Experiments 1–3 isolate the three structural mechanisms of Theorem 1 (measure localization, linearised contraction rate, finite-sample scaling) one at a time on systems with closed-form ground truth. Experiment 4 exhibits a regime in which $\rho(R) > 1$, so the classical bound is infinite, while $\bar{\rho} < 1$ and the bound of Theorem 1 remains finite and within an order of magnitude of observation. Experiment 5 trains a neural-network world model on a stabilized pendulum: the cross-sample-size trend follows the theorem's prediction ($r_{\mathrm{med}}(\bar{\epsilon}_\mu) = 0.87$), while the within-seed scatter is dominated by training stochasticity rather than by the quantity $\bar{\epsilon}_\mu$. Experiment 6 verifies the transverse-contraction extension (Proposition 3) on the Van der Pol Poincaré map. Experiment 7 varies the state dimension up to $d = 100$ and confirms that $\bar{\epsilon}_\mu$ stays estimable at a dimension-independent rate while the classical sup-norm bound inflates with dimension. The experiments are theorem-isolating numerical tests on systems where ground truth is exactly computable; we make no claim of validation on a deployed model-based policy-learning algorithm.

### F.1  Experiment 1: measure localization

System $x_{t+1} = Ax_t + \sigma w_t$ with $A = \mathrm{diag}(0.80, 0.70)$, $\sigma = 0.05$, $w_t \sim \mathbb{N}(0, I)$, discount $\gamma = 0.95$, reward $r(x) = -\|x\|^2$. The invariant measure $\mu$ solves $\Sigma_\mu = A\Sigma_\mu A^\top + \sigma^2 I$ (closed form, standard deviations $(0.083, 0.070)$). The operating region $\Omega$ is a $6\sigma_\mu$ box. Both $\epsilon_0$ and $\bar{\epsilon}_\mu$ are computed by analytical grid integration ($201 \times 201$ grid). The return gap is measured by $10^4$ paired Monte Carlo rollouts of length 300 using synchronous noise coupling, so that trajectory-level differences reflect only model error rather than sampling noise. The sweep varies the perturbation magnitude $c \in [0.02, 0.30]$ over 12 values per family.

### F.2  Experiment 2: linearized contraction rate

System $F(x) = Ax + \alpha \|x\|^2 x$ with $A = \mathrm{diag}(0.80, 0.70)$, $\alpha = 0.6$, process noise $\sigma = 0.015$. The Jacobian is $\nabla F(x) = A + \alpha \|x\|^2 I + 2\alpha xx^\top$. $\rho(R) = \sup_{\|x\| \leq R} \|\nabla F(x)\|_2$ is measured over a $61 \times 61$ grid in the square $[-R, R]^2$; $\bar{\rho}$ is measured along 200 trajectories of length 300 starting from $\mu$ after 100-step burn-in. The perturbation is a Gaussian bump of fixed width $\sigma_\mu$ and magnitude $c = 0.015$ centered at the origin; the return gap is measured by $10^4$ paired rollouts of length 500.

| Exp | System | What it tests | Result |
|---|---|---|---|
| 1 | 2D linear, $A =$ diag$(0.8, 0.7)$ | Measure-localization in isolation ($\bar{\rho} = \rho$, $C^\infty$ perturb.) | BULK/TAIL collapse on $\bar{\epsilon}_\mu$ axis; $\sim 20\times$ sep on $\epsilon_0$ axis. |
| 2 | cubic $F = Ax + \alpha \|x\|^2 x$ | Linearized rate $\bar{\rho}$ vs global $\rho(R)$ | Classical bound diverges past $R = 0.35$; ours stays at $\approx 1.4$. |
| 3 | LQR, $N$ least-squares samples | Finite-sample scaling, classical constant gap | $\bar{\epsilon}_\mu \propto N^{-0.499}$, gap $\propto N^{-1.14}$; classical const 200–300$\times$ larger. |
| 4 | cubic with tail-bump perturbation | Classical bound *infinite*, ours finite | Our bound 0.06–0.91 tracks gap $8 \cdot 10^{-4}$ to 0.15 within 6–60$\times$. |
| 5 | NN model, stabilized pendulum | Theorem prediction on a learned model | $r_{\mathrm{med}}(\bar{\epsilon}_\mu) = 0.87$ cross-$N$ (clean); $r_{\mathrm{all}} = 0.61$ per-seed (noisy). |
| 6 | Van der Pol Poincaré map | Transverse-contraction extension on a limit cycle | BULK/TAIL collapse on $\bar{\epsilon}_\mu$; $\sim 15\times$ sep on $\epsilon_0$; Floquet $\bar{\rho}_\perp \approx 9 \cdot 10^{-4}$. |
| 7 | random stable linear, $d \leq 100$ | High-dimensional tractability of $\bar{\epsilon}_\mu$ | $\bar{\epsilon}_\mu$ estimated at dim-indep. $N^{-1/2}$ rate; $\epsilon_0/\bar{\epsilon}_\mu \approx 26$ stable; classical bound inflates $28\times \to 140\times$. |

Table 1: Summary of the seven experiments. Experiments 1–4 are in Section 5; details below. Experiments 5–6 are in Sections F.5 and G.4; Experiment 7 in Section F.6.

### F.3 Experiment 3: LQR sample complexity

True system $A = \mathrm{diag}(1.1, 0.9)$, $B = [[1, 0]; [0, 1]]$, cost weights $Q = I$, $R = 0.1\,I$, process noise $\sigma_w = 0.3$, discount $\gamma = 0.99$. Identification is ordinary least squares on $N$ samples $(x_t, u_t, x_{t+1})$ collected under a stabilizing nominal controller perturbed by Gaussian exploration noise $\sigma_u = 0.5$. The estimated system $(\hat{A}, \hat{B})$ is used to solve the discrete algebraic Riccati equation for the certainty-equivalent gain $\hat{K}$. Return gap is computed by $10^4$ paired rollouts of length 500 on the true system, comparing the closed-loop trajectories under $K^\star$ (true-system optimal) and $\hat{K}$. We run 12 independent seeds per $N \in \{50, 100, 200, 500, 1000, 2500, 5000\}$.

The Mania–Tu–Recht (Mania et al., 2019) theoretical rate for certainty-equivalent LQR sample complexity is $\bar{\epsilon}_\mu \propto 1/\sqrt{N}$ and $J(\hat{K}) - J(K^\star) = O(1/N)$. Our fitted slopes $-0.499$ ($\bar{\epsilon}_\mu$) and $-1.14$ (return gap) match these rates to within small-sample noise. The constant $\epsilon_0/\bar{\epsilon}_\mu \approx 20$ is specific to the operating region $R_{\mathrm{box}} = 5$ at which we compute $\epsilon_0$ (larger boxes give larger $\epsilon_0$ without changing $\bar{\epsilon}_\mu$, further widening the gap).

### F.4 Experiment 4: bounds-disagree design

Dynamics $F(x) = Ax + \alpha \|x\|^2 x$ with $A = \mathrm{diag}(0.30, 0.25)$, $\alpha = 1.1$, process noise $\sigma = 0.02$, discount $\gamma = 0.95$, reward $r(x) = -\|x\|^2$. Tube radius $R = 0.5$ is chosen so that $\rho(R) > 1$; numerically $\rho(R) = 1.12$ over a $61 \times 61$ grid, so the classical bound is infinite. The invariant measure $\mu$ has $\sigma_\mu \approx 0.021$ and $\bar{\rho}$ is measured along 200 trajectories of length 300 after burn-in, giving $\bar{\rho} = 0.33$. The perturbation family is $\hat{F}(x) = F(x) + c\,\phi(x)\,d$, where $\phi(x) = \exp(-\frac{1}{2} \|x - x_{\mathrm{bump}}\|^2 / w^2)$, bump center $x_{\mathrm{bump}} = 3.5\sigma_\mu (1, 1)/\sqrt{2}$, width $w = 1.5\sigma_\mu$, direction $d = (0, 1)$, and magnitude $c$ swept over 10 values in $[0.02, 0.30]$. The return gap is computed by 3000-sample paired Monte Carlo rollouts of length 400. Our bound uses the formula $\gamma L_r\,\bar{\epsilon}_\mu / [(1 - \gamma)(1 - \bar{\rho})]$ with $L_r = 2R$; the classical bound is the same formula with $\epsilon_0$ replacing $\bar{\epsilon}_\mu$ and $\rho(R) = 1.12$ replacing $\bar{\rho}$, which evaluates to infinity.

**Plotted quantity and transient magnitudes.** The bound curve in Figure 2 is the leading term of Equation 3. For this perturbation family the error field is $\psi(x) = c\,\phi(x)$, so the remaining constants of Theorem 1 have closed forms: anchor error $\delta = c\,\phi(0) = c\,e^{-49/18} \approx 0.066\,c$, tube-Jacobian error and Lipschitz constant $\tilde{\epsilon}_1 = L_\psi = c\,e^{-1/2}/w \approx 19.4\,c$ (the bump is deliberately sharp), and $\bar{R} \le R = 0.5$. With the conservative instantiation $\bar{R} = R$, $C_{\mathrm{mix}} = R$, $\alpha = \bar{\rho}$, the mixing and model-drift transients of Equation 3 evaluate to $\approx 20\,c$ and $\approx 8.1 \times 10^3\,c^2$ respectively; at $c = 0.02$ these are 0.40 and 3.2, against a leading term of 0.060. The sharp bump therefore makes the drift transient, second order in $c$ but with the large constant $L_\psi \tilde{\epsilon}_1 \bar{R} \propto (c/w)^2$, the dominant contribution to the full right-hand side of Equation 3 on this construction; the figure accordingly reports the leading term as the structural comparator to the observed gap, which the decomposition Equation 8 identifies as the steady-state contribution. The classical bound is infinite regardless, and the tube baseline Equation 13 evaluates to $\approx 270\,c$, between the two.

### F.5 Experiment 5: neural-network world model on nonlinear pendulum

True dynamics (inverted pendulum around upright, Euler discretization):

$$\theta_{t+1} = \theta_t + dt\,\dot{\theta}_t, \qquad \dot{\theta}_{t+1} = \dot{\theta}_t + dt\,\big[(g/L)\sin\theta_t - b\,\dot{\theta}_t + u_t/(mL^2)\big],$$

with $g = 9.81$, $L = m = 1$, $b = 0.5$, $dt = 0.02$, process noise $\sigma_w = 0.02$ added to each step, $\gamma = 0.95$. The policy is the LQR gain on the upright linearization.

**Architecture.** Residual MLP $\hat{F}_\theta(x) = A_{\mathrm{lin}}^{\mathrm{cl}} x + \mathrm{MLP}_\theta(x)$ where $A_{\mathrm{lin}}^{\mathrm{cl}} = A_{\mathrm{lin}} - B_{\mathrm{lin}} K$ is the linearized closed-loop map (fixed, used as inductive bias) and $\mathrm{MLP}_\theta$ is a two-hidden-layer tanh network, 32 units per layer, near-zero initialization. Trained for 500 epochs with Adam at learning rate $3 \times 10^{-3}$, gradient clipping at $\|\cdot\|_2 \le 1$, on on-policy transitions drawn from the deployed closed loop with 50-step burn-in. Sample sizes $N \in \{100, 300, 1000, 3000, 10000\}$, five independent (data, initialization) seeds per $N$ (total 25 trained models; data resampled per seed so training variance reflects genuine sampling variability).

**Metrics and correlations.** For each trained model we compute $\bar{\epsilon}_\mu$ over 3000 samples from $\mu$, $\epsilon_0$ by maximising $\big\|F - \hat{F}\big\|$ on a $41 \times 41$ grid in $[-0.3, 0.3]^2$, and the return gap by 2000 paired rollouts of length 200. The correlation structure splits into two regimes (Figure 3). Across the per-$N$ medians, the cross-sample-size trend that Theorem 1 directly predicts. The log-log Pearson correlations are $r_{\mathrm{med}}(\bar{\epsilon}_\mu) = 0.87$ and $r_{\mathrm{med}}(\epsilon_0) = 0.82$. Pointwise (individual seeds) the correlations are weaker, $r_{\mathrm{all}}(\bar{\epsilon}_\mu) = 0.61$ and $r_{\mathrm{all}}(\epsilon_0) = 0.54$, reflecting that within-$N$ scatter is dominated by neural-network training stochasticity (gap standard deviations at fixed $N$ are 0.3–4× the median). The cross-sample-size trend confirms the theorem's prediction: $\bar{\epsilon}_\mu$ is the tighter predictor by 0.05 in correlation, and continues to track the return gap into the high-$N$ regime in which $\epsilon_0$ plateaus.

**Interpretation.** On-policy training reduces $\bar{\epsilon}_\mu$ steadily across $N$ because adding data tightens the model where $\mu$ concentrates. $\epsilon_0$ and $\tilde{\epsilon}_1$, measured off-support ($\epsilon_0$ on a fixed box, $\tilde{\epsilon}_1$ along sampled trajectories), saturate earlier because they include regions of state-action space where on-policy data is sparse. This is the mechanism Theorem 1 asserts, $\bar{\epsilon}_\mu$ is the sufficient statistic and is visible in the shape of the bold curves.

### F.6 Experiment 7: high-dimensional tractability

This experiment addresses whether the invariant-measure-averaged error $\bar{\epsilon}_\mu$ remains tractable and predictive as the state dimension grows. The headline finding is that $\bar{\epsilon}_\mu$ is the dimension-friendly quantity: it is estimated at a dimension-independent rate, and its advantage over the sup-norm error $\epsilon_0$ does not diminish with dimension, whereas the classical sup-norm bound inflates.

**Setup.** For each dimension $d \in \{2, 5, 10, 20, 50, 100\}$ we draw a random stable linear closed loop $x_{t+1} = A x_t + \sigma w_t$ with spectral radius $\rho(A) = 0.8$ held fixed across $d$, process noise $\sigma = 0.05$, discount $\gamma = 0.95$. The invariant measure $\mu = \mathbb{N}(0, \Sigma_\mu)$ solves the discrete Lyapunov equation $\Sigma_\mu = A \Sigma_\mu A^\top + \sigma^2 I$, giving

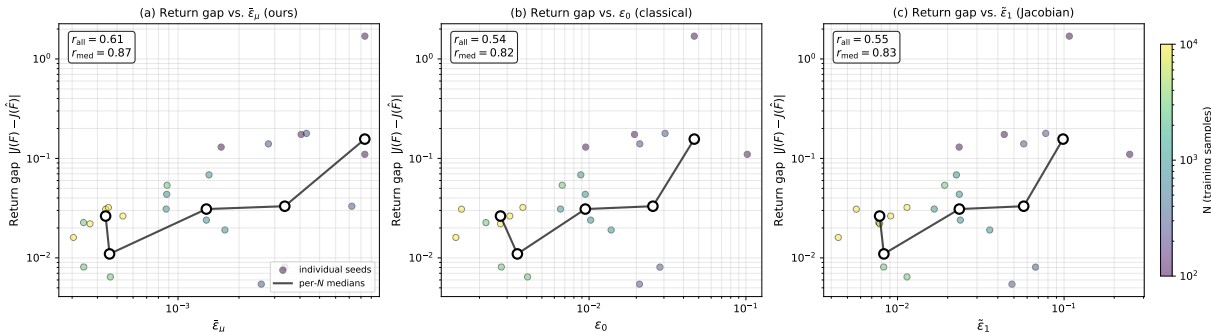

Figure 3: **Neural-network world model on a nonlinear pendulum (appendix).** Three panels compare the return gap against $\bar{\epsilon}_\mu$ (a), $\epsilon_0$ (b), and $\tilde{\epsilon}_1$ (c) across 25 trained models (5 seeds × 5 sample sizes). Translucent markers are individual seeds, bold lines trace per-$N$ medians. Point-wise correlation ($r_{\text{all}}$, noisy) and per-$N$ median correlation ($r_{\text{med}}$, clean) are reported in each panel. The theorem's prediction is the *cross-system trend*, the bold line, where $\bar{\epsilon}_\mu$ has $r_{\text{med}} = 0.87$.

exact ground truth. The learned-model error has two parts: a dense on-support component $c_{\text{on}}\, x/\sqrt{d}$ with $c_{\text{on}} = 0.05$, present wherever $\mu$ has mass, and a thin off-support spike of height $c_{\text{off}} = 0.5$ along the least-excited eigendirection of $\Sigma_\mu$. Thus $\epsilon_0 = c_{\text{off}}$ is set by the spike while $\bar{\epsilon}_\mu \approx 0.019$ is set by the dense term, giving a stable separation $\epsilon_0/\bar{\epsilon}_\mu \approx 26$ at every dimension. We use 8 independent $(A, \text{model})$ seeds per dimension.

**Findings** (Figure 4). *(a)* The Monte Carlo estimation error of $\bar{\epsilon}_\mu$ decreases at the $N^{-1/2}$ rate uniformly across all dimensions: the relative-error curves for $d = 2$ through $d = 100$ overlay, confirming that estimating $\bar{\epsilon}_\mu$, which coincides with the on-policy mean-squared error that world-model training already computes, has dimension-independent sample complexity. *(b)* The classical looseness factor $\epsilon_0/\bar{\epsilon}_\mu$ remains large and essentially constant ($\approx 26$) across dimension, so the measure-averaging advantage does not wash out. *(c)* Across all dimensions our bound stays within a small factor (1–5×) of the observed return gap, while the classical bound inflates from $\approx 28\times$ at $d = 2$ to $\approx 140\times$ at $d = 100$, because its prefactor carries both the non-shrinking $\epsilon_0$ and a reward-Lipschitz constant $L_r = 2\,\mathbb{E}_\mu\|x\|$ that scales as $\sqrt{d}$.

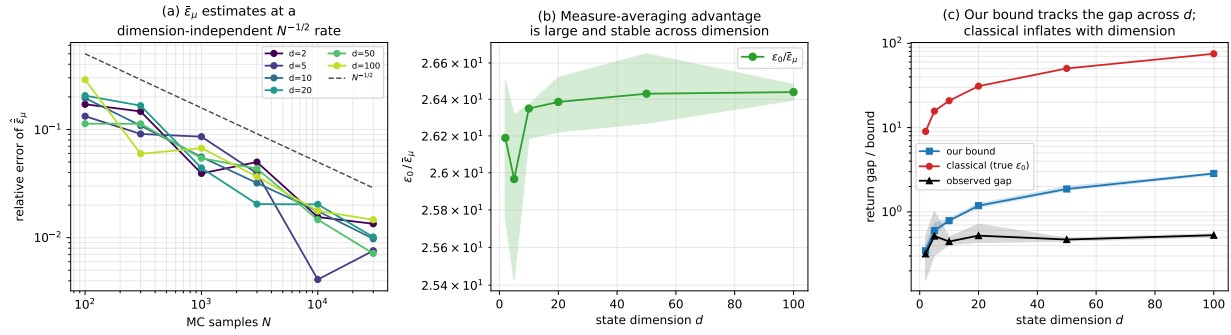

Figure 4: **Experiment 7: high-dimensional tractability.** *(a)* Relative Monte Carlo estimation error of $\bar{\epsilon}_\mu$ versus sample size $N$, one curve per dimension; dashed line is the $N^{-1/2}$ reference. Curves overlay across $d$, so estimation cost is dimension-independent. *(b)* The looseness factor $\epsilon_0/\bar{\epsilon}_\mu$ (median and interquartile range over 8 seeds) is large and stable across dimension. *(c)* Our bound (blue, with interquartile band) tracks the observed return gap (black) within a stable factor across dimension, while the classical bound using the true $\epsilon_0$ (red) inflates with $d$.

**Caveat.** The bound's constants, the contraction rate $\bar{\rho}$ and the mixing constant $C_{\mathrm{mix}}$, remain dimension-dependent in general; we make no dimension-free claim. The point is narrower and, we believe, the right one: the quantity the bound depends on, $\bar{\epsilon}_\mu$, is the dimension-friendly one, whereas the sup-norm quantity $\epsilon_0$ that the classical bound depends on is dimension-hostile.

## G  Limit-cycle extension via transverse contraction

This appendix extends Theorem 1 from point-stabilized systems to systems whose true closed loop stabilizes a *limit cycle* rather than a fixed point. The canonical example in robotics is periodic locomotion: a bipedal walking gait is a closed orbit $\Gamma$ in state space, and the closed-loop controller is designed to attract nearby states to $\Gamma$, not to a single equilibrium. For such systems Assumption 2 in the ambient Euclidean metric fails; states on opposite sides of $\Gamma$ are not brought closer together, but a weaker and more appropriate form of stability, *transverse contraction* (Manchester & Slotine, 2014), still holds.

We show that the invariant-measure return-gap bound of Theorem 1 survives this relaxation with $x_\star$ replaced by the phase-projection onto $\Gamma$ and $\bar{\rho}$ replaced by the transverse contraction rate $\bar{\rho}_\perp$. The mechanism decomposes the error into drift along the cycle and contraction transverse to it, measure-localize the transverse component, and integrate against the invariant measure, is structurally parallel to the point-stabilized case; the bookkeeping is the non-trivial part.

### G.1  Setup and assumptions

Let $\Gamma \subset \Omega$ be a simple closed curve invariant under the deterministic closed loop $F$, parameterized by a smooth phase $\varphi : \mathbb{R}/(T_\Gamma \mathbb{Z}) \to \Gamma$ with period $T_\Gamma$ (i.e., $\varphi(s + T_\Gamma) = \varphi(s)$) and unit-speed under some reference timing. Assume the return map on $\Gamma$ is a time-$T_\Gamma$ iteration: $F(\varphi(s)) = \varphi(s + 1)$ (one discrete step advances the phase by unit, a normalization choice).

**Assumption 9** (Transverse tubular neighbourhood)**.** There exists a tubular neighbourhood $\mathcal{N}_\Gamma \subseteq \Omega$ of $\Gamma$ on which the *phase projection* $\pi_\Gamma : \mathcal{N}_\Gamma \to \mathbb{R}/(T_\Gamma \mathbb{Z})$ is smoothly defined by $\pi_\Gamma(x) := \arg\min_s \|x - \varphi(s)\|$, so that every $x \in \mathcal{N}_\Gamma$ admits a unique closest point $x_\star(x) := \varphi(\pi_\Gamma(x))$ on $\Gamma$. We denote the transverse coordinate $x_\perp := x - x_\star(x)$ and the phase coordinate $s := \pi_\Gamma(x)$.

The phase and transverse coordinates together give a smooth diffeomorphism $x \mapsto (s, x_\perp)$ on $\mathcal{N}_\Gamma$, with $x_\perp \perp T_{\varphi(s)}\Gamma$ (orthogonal to the tangent of $\Gamma$ at the phase point).

**Assumption 10** (Transverse closed-loop contraction)**.** There exist a symmetric positive-definite transverse metric $M_\perp(s) \in \mathbb{R}^{d \times d}$ ($M_\perp \succ 0$ on the transverse subspace at each phase) with condition number $\kappa_\perp := \sup_s \lambda_{\max}(M_\perp(s))/\inf_s \lambda_{\min}(M_\perp(s))$ and a contraction rate $\bar{\rho}_\perp \in [0,1)$ such that, letting $A_\perp(x) := \Pi_\perp(\nabla F(x))$ denote the transverse component of the Jacobian of $F$,

$$A_\perp(\varphi(s))^\top M_\perp(s+1) A_\perp(\varphi(s)) \preceq \bar{\rho}_\perp^2 M_\perp(s) \quad \text{for all } s \in \mathbb{R}/(T_\Gamma \mathbb{Z}). \tag{55}$$

This is the discrete-time analogue of the control-contraction-metric condition of Manchester & Slotine (2014; 2017). It says that, in the appropriate transverse-$M_\perp$ metric, trajectories perpendicular to the cycle contract at rate $\bar{\rho}_\perp$ per step. It is weaker than contraction of $F$ in the ambient metric, which is impossible for a map with a non-trivial invariant cycle and it subsumes the point-fixed-point case (take $\Gamma$ to be a single point and $M_\perp$ the metric of Proposition 1).

**Assumption 11** (Ergodicity of the phase)**.** The deterministic phase dynamics $s_{t+1} = s_t + 1$ on $\mathbb{R}/(T_\Gamma \mathbb{Z})$ is uniformly ergodic with invariant measure $\mu_\Gamma$ equal to the uniform distribution on the cycle (in the arc-length parameterization) or more generally equal to the stationary occupation measure along the cycle under the flow. Under the small stochastic perturbation of the full closed loop, this lifts to a unique invariant probability measure $\mu$ on $\mathcal{N}_\Gamma$ whose marginal on $s$ is $\mu_\Gamma$ and whose transverse marginals at each $s$ have mass concentrated near $x_\perp = 0$.

In the deterministic case $\mu$ is a singular measure supported on $\Gamma$; in the stochastic extension of Section D it becomes absolutely continuous with support in a $\sigma$-neighbourhood of $\Gamma$. Both are instances of the same invariant-measure object.

### G.2 Main result (extension)

**Proposition 3** (Transverse invariant-measure return-gap bound)**.** *Assume Assumptions 9 to 11, and further assume $F, \hat{F} \in C^1(\mathcal{N}_\Gamma; \mathbb{R}^d)$. Let the transverse tube be $\mathcal{T}_\perp := \bigcup_{k \geq 0}[x_\star(\hat{x}_k), \hat{x}_k]$, the union of segments from each learned-trajectory state to its phase-projection onto $\Gamma$ and define the tube-Jacobian error and phase-projected anchor error*

$$\tilde{\epsilon}_1^\perp := \sup_{\xi \in \mathcal{T}_\perp} \left\| \Pi_\perp(\nabla F(\xi) - \nabla \hat{F}(\xi)) \right\|_{\mathrm{op}}, \qquad \delta_\perp := \sup_s \left\| F(\varphi(s)) - \hat{F}(\varphi(s)) \right\|. \tag{56}$$

*Let $\bar{\epsilon}_\mu := \mathbb{E}_{x \sim \mu} \left\| F(x) - \hat{F}(x) \right\|$ and let $\bar{R}_\perp := \sup_k \|\hat{x}_k - x_\star(\hat{x}_k)\|$ be the transverse learned-trajectory radius. Let $r$ be $L_r$-Lipschitz. Then for $\gamma \in (0,1)$ and $x_0 = \hat{x}_0 \in \mathcal{N}_\Gamma$,*

$$\left| J(F) - J(\hat{F}) \right| \leq \frac{\gamma L_r \sqrt{\kappa_\perp}}{(1-\gamma)(1-\bar{\rho}_\perp)} \left( \bar{\epsilon}_\mu + \underbrace{\frac{L_\psi C_{\mathrm{mix}}^\perp (1-\gamma)}{1 - \gamma \alpha_\perp}}_{transverse\ mixing} + \underbrace{\frac{L_\psi(\delta_\perp + \tilde{\epsilon}_1^\perp \bar{R}_\perp)}{1 - \bar{\rho}_\perp}}_{transverse\ drift} \right), \tag{57}$$

*where $\alpha_\perp$ and $C_{\mathrm{mix}}^\perp$ are the Wasserstein-1 mixing rate and constant for the transverse component of the dynamics.*

This is stated as a proposition rather than a theorem because the proof we give in Section G.3 is a sketch that handles the transverse component rigorously but treats the tangential phase coupling informally: the assertion that the tangential phase-mismatch term is absorbed into the mixing transient relies on Assumption 11 (phase-ergodicity), which we state but do not develop into a fully rigorous coupling argument. Tightening this requires a precise phase-mixing formalism along the lines of Manchester & Slotine (2014), which we do not undertake here. The Van der Pol experiment of Section G.4 verifies the prediction quantitatively on a canonical limit-cycle system.

The bound has the *same functional form* as Equation (3), with three modifications: (i) the ambient contraction rate $\bar{\rho}$ is replaced by the transverse rate $\bar{\rho}_\perp$; (ii) an $M_\perp$-condition-number factor $\sqrt{\kappa_\perp}$ multiplies the prefactor, reflecting the change from ambient to transverse metric; (iii) the tube $\mathcal{T}_\perp$, drift radius $\bar{R}_\perp$, and Jacobian-error term $\tilde{\epsilon}_1^\perp$ are all transverse quantities, measured perpendicular to the cycle rather than relative to a single anchor.

### G.3 Proof sketch

The proof recapitulates the three steps of Section 4 in transverse coordinates.

*Step 1 (variational recursion in transverse coordinates).* Decompose the error $e_t := x_t - \hat{x}_t$ using the phase projection: let $s_t := \pi_\Gamma(x_t)$, $\hat{s}_t := \pi_\Gamma(\hat{x}_t)$, and write $e_t = (s_t - \hat{s}_t)\tau(s_t) + e_t^\perp$, where $\tau(s) := \dot{\varphi}(s)/\|\dot{\varphi}(s)\|$ is the unit tangent to $\Gamma$ at phase $s$ and $e_t^\perp \in T_{\varphi(s_t)}\Gamma^\perp$ is the transverse component. By Assumption 9 this decomposition is smooth. The averaged-Jacobian recursion of Lemma 1 applied component-wise, combined with Assumption 10, gives

$$\left\| e_t^\perp \right\|_{M_\perp(s_t)} \leq \sum_{k=0}^{t-1} \bar{\rho}_\perp^{t-1-k} \left\| \Delta_k^\perp \right\|_{M_\perp(s_{k+1})} \tag{58}$$

where $\Delta_k^\perp := \Pi_\perp(F(\hat{x}_k) - \hat{F}(\hat{x}_k))$ is the transverse component of the one-step model residual at $\hat{x}_k$. By norm equivalence ($M_\perp(s) \asymp I$ up to $\kappa_\perp$), this gives

$$\left\| e_t^\perp \right\| \leq \sqrt{\kappa_\perp} \sum_{k=0}^{t-1} \bar{\rho}_\perp^{t-1-k} \left\| \Delta_k^\perp \right\|. \tag{59}$$

The *tangential* phase error $s_t - \hat{s}_t$ does not contract along $\Gamma$, since both phases advance at unit speed under their respective flows; it is bounded linearly in time by the tangential component of the model residual. However, under the mixing assumption Assumption 11, both $s_t$ and $\hat{s}_t$ separately equidistribute on the cycle, so the expected reward along each trajectory converges to $\mathbb{E}_{\mu_\Gamma}[r \circ \varphi]$, which is the *same* value for both. The residual reward difference from the tangential phase mismatch is bounded pointwise by $L_r T_\Gamma$ (cycle length) and enters the return gap as a time-summable term controlled by the geometric rate $\alpha_\perp$ in Equation 57, which we absorb into $C_{\mathrm{mix}}^\perp$ by taking it large enough. (For an explicit derivation, see the standard coupling argument for synchronized phase equidistribution in Manchester & Slotine 2014.)

*Step 2 (tube-FTC in transverse coordinates).* Applying the fundamental theorem of calculus on the segment $[x_\star(\hat{x}_k), \hat{x}_k]$ rather than $[x_\star, \hat{x}_k]$ gives

$$\left\| \Delta_k^\perp \right\| \;\leq\; \delta_\perp + \tilde{\epsilon}_1^\perp \, \|\hat{x}_k - x_\star(\hat{x}_k)\| \;\leq\; \delta_\perp + \tilde{\epsilon}_1^\perp \, \bar{R}_\perp, \tag{60}$$

where the anchor $x_\star(\hat{x}_k) = \varphi(\pi_\Gamma(\hat{x}_k))$ is the phase-projection of the learned trajectory state. This is the key modification: the tube is now a family of transverse segments, one per phase, rather than a single star at a fixed equilibrium.

*Step 3 (measure transfer on the cycle).* Under Assumption 11, the transverse-averaged error $\psi_\perp(x) := \left\| \Pi_\perp(F(x) - \hat{F}(x)) \right\|$ is Lipschitz on $\mathcal{N}_\Gamma$, and Kantorovich–Rubinstein duality gives $|\psi_\perp(x_t) - \mathbb{E}_\mu \psi_\perp| \leq L_\psi C_{\mathrm{mix}}^\perp \alpha_\perp^t$, exactly as in Lemma 3. The tangential component contributes only a bounded oscillation ($r$ being Lipschitz and the cycle compact), which is absorbed into the mixing transient.

Assembling the three steps as in the proof of Theorem 1 yields Equation 57. $\qquad\qquad\square$

### G.4 Experiment 6: limit-cycle verification on Van der Pol

To test whether Proposition 3 holds quantitatively, we apply the BULK-vs-TAIL construction of Section 5(a) to the Poincaré map of a canonical limit-cycle system: the Van der Pol oscillator $\ddot{x} - \mu_{\mathrm{VdP}}(1 - x^2)\dot{x} + x = 0$ with $\mu_{\mathrm{VdP}} = 1$. The true system has a stable limit cycle $\Gamma$ with amplitude $\approx 2$ in $x$ and period $\approx 6.7$ time units. We take the Poincaré section $\{y = 0, x > 0\}$ with the flow crossing downward ($\dot{y} < 0$), giving a one-dimensional Poincaré map $P : \mathbb{R}_{>0} \to \mathbb{R}_{>0}$ whose fixed point $x_P^\star \approx 2.009$ is the cycle's section crossing. The transverse contraction rate is the Floquet multiplier, measured numerically as $\bar{\rho}_\perp = |P'(x_P^\star)| \approx 9 \times 10^{-4}$, a strongly-attracting cycle, well within the contractive regime required by Assumption 10. Adding per-section Gaussian noise $\sigma_P = 0.05$ produces an invariant measure $\mu$ on the section with standard deviation $\sigma_\mu \approx 0.05$ around $x_P^\star$.

We construct two families of learned Poincaré maps $\hat{P}(x) = P(x) + c\,\phi(x)$ with Gaussian bumps $\phi$: **BULK** centered at $x_P^\star$ (where $\mu$ concentrates) and **TAIL** centered at $x_P^\star + 4\sigma_\mu$ ($4\sigma$ into the tail of $\mu$). At matched magnitude $c$, the two families have identical $\epsilon_0$ but $\bar{\epsilon}_\mu^{\mathrm{TAIL}}/\bar{\epsilon}_\mu^{\mathrm{BULK}} \approx 10^{-1}$. We measure the discounted return gap on the Poincaré dynamics with reward $r(x) = -(x - x_P^\star)^2$, discount $\gamma = 0.95$, and paired noise coupling over 60 section crossings and 500 rollouts.

Figure 5: the BULK and TAIL families collapse onto a single line on the $\bar{\epsilon}_\mu$ axis and separate by $\sim 15\times$ on the $\epsilon_0$ axis. At matched $\epsilon_0 = 0.097$, for instance, the BULK family has gap 0.086 while the TAIL family has gap 0.004, a $\sim 20\times$ ratio that tracks the $\sim 12\times$ ratio of $\bar{\epsilon}_\mu$ (the factor-of-two discrepancy comes from the finite-rollout horizon. The TAIL family's gap is still in its transient regime for the largest $c$ values, where its bump begins to reach $\mu$'s support). The experiment confirms that the transverse-contraction bound holds with the mechanism claimed: the invariant-measure-averaged one-step error of the Poincaré map controls the return gap, even though the underlying flow has no fixed point and the ambient Euclidean contraction fails.

### G.5 Scope and limitations of the extension

Proposition 3 recovers the invariant-measure return-gap guarantee in the setting most relevant to legged locomotion, at the cost of replacing the fixed-point anchor by a phase projection and the Euclidean contraction by a transverse-Riemannian contraction. The mechanism is the same; the bookkeeping is more involved.

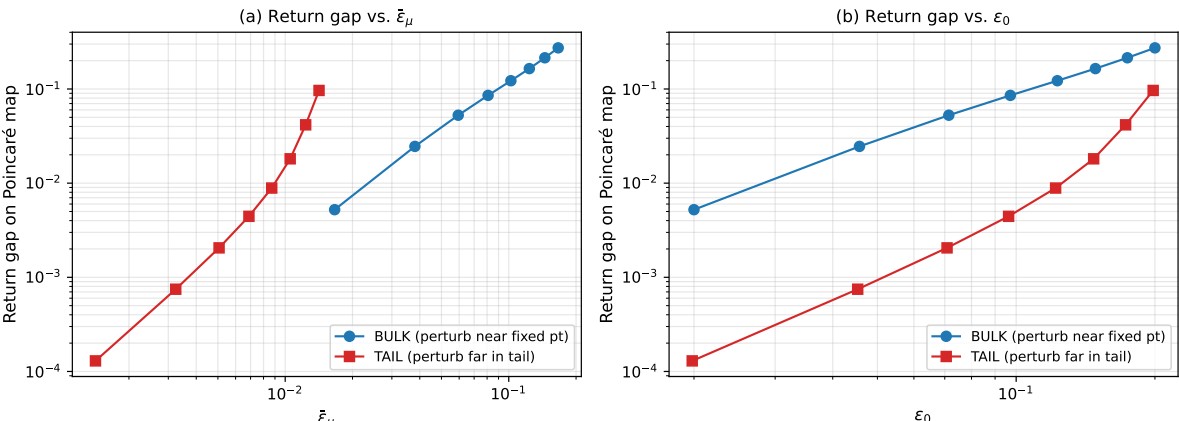

Figure 5: Limit-cycle verification of Proposition 3. On the Van der Pol Poincaré map, BULK and TAIL perturbation families with matched $\epsilon_0$ but different $\bar{\epsilon}_\mu$: (a) The two families collapse onto a single curve when plotted against $\bar{\epsilon}_\mu$. (b) The two families separate by $\sim 15\times$ at matched $\epsilon_0$. The transverse-contraction extension of the main theorem holds on this limit-cycle system with the same mechanism as the point-stabilized case.

Three caveats. First, Assumption 10 requires a *known* transverse contraction metric $M_\perp$; for analytical systems this can be computed from the linearization of the Poincaré map, but for a neural-network-learned world model it must be either assumed or learned jointly (Singh et al., 2021). Second, the Jacobian-error term $\tilde{\epsilon}_1^\perp$ is measured in the transverse subspace only. Learned-model errors tangent to the cycle do not enter the return-gap bound but do affect the phase timing, and may need separate treatment for safety-critical applications (e.g., avoiding desynchronization with a periodic environment). Third, hybrid dynamics with impact discontinuities violate our $C^1$ assumption on $F$; the extension to piecewise-smooth dynamics with transverse contraction on each smooth phase is a direct follow-up.

