# OpenReview forum: "On-Policy Model Error Suffices:  An Invariant-Measure Return-Gap Bound for Model-Based Reinforcement Learning"
_TMLR — Under review for TMLR_

### Review · Reviewer_Eb3m · 2026-05-18

**Summary Of Contributions:**

This paper studies a model-based framework where the performance $J$ varies by the true model $F$ and a learned model $\hat{F}$. The authors adopt a bounded state space and a closed-loop stable dynamics. Assumption 4 seems to be the key hypothesis where the local first-order and a scalar error on a tube are introduced. The authors also rely on the forward invariance assumption. The authors argue an improved error bound using invariant measure--which replaces the conventional sup-norm error bound. Several small toy examples support their theoretical results.


*Strength*

1. The paper provides a new bound error bound  compared to the previous literature. In particular, the paper proves a invariant-measure error bound compared to previous sup-norm error bound $\sup_x ||F(x)-F(\hat{x})||$ with $E_{\mu} [||F(x)-F(\hat{x}) || ]$. These two are clearly different bounds, and the authors manage to derive such bound. Theorem 6 provides this result.

2. The framework is not limited to convergence toward a single equilibrium point. The paper further discusses extensions to limit-cycle systems through transverse contraction analysis.

*Weakeness*

1. The paper requires strict assumptions including the contraction condition and the forward invariance condition. It is not clear how standard the Assumption 4--forward invariance--is n the literature. Moreover, for a contractive system, .e.g., stable linear system, we may easily find a forward invariance set that depends on an initial condition . Why is this assumption necessary?

2. Even though the error bound is interesting, the practical significance of the bound is questionable. In particular, it is not evident whether the improved invariant-measure-based bound would remain meaningful or tractable in high-dimensional settings

**Audience:**

Yes

**Audience Explanation:**

Model-based RL is an important topic in the RL community. Understanding the error propagation of learned models is a fundamental problem. In this regard, the paper provides a new type of error bound under several structural assumptions, which may be of interest to the community.

**Claims And Evidence:**

Yes

**Claims Explanation:**

The main theorem of the paper (Theorem 6) looks solid. The authors provide a sketch of proof and the technical lemmas in Section 4.2 support the theorem which are clearly explained.

**Requested Changes:**

1. The introduction part reads more like a compressed theorem statement than an intuition-building introduction. The authors are using technical control-theoretic terminology before giving intuition, which makes the first two paragraphs too dense.

2. There is a typo in Theorem 6 : Theorem 1-5 -> Assumption 1-5.

3. In section 4, the authors directly provide a sketch of proof--which is again difficult to grasp what the authors are trying to eventually doing. In particular, it is unclear what quantity the authors are ultimately trying to transform and why the decomposition in equation (8) is the natural object to consider.

4. The experiments seem to have stochastic noise. However, Figure 1 and 2 do not report standard deviations, confidence intervals, or error bars.

---

> ### Author Response · Authors · 2026-07-07
> **Part 1: Addressing weaknesses**
>
> We thank the reviewer for the careful reading, the positive assessment on both TMLR criteria, and the useful requests. A revised PDF has been uploaded, with substantive changes marked in color. Below we quote each point and give the corresponding change,
>
> ## Weakness 1
> > "It is not clear how standard the Assumption 4 -- forward invariance -- is in the literature. Moreover, for a contractive system, e.g., stable linear system, we may easily find a forward invariance set that depends on an initial condition. Why is this assumption necessary?"
>
> The assumption is standard; it is necessary because it must hold for the *learned* model $\hat F$, which the automatic construction the reviewer describes does not cover; and for contractive systems it is mild in exactly the way the reviewer suggests, which Lemma A below makes formal.
>
> **How standard.** Positively invariant operating regions are a cornerstone hypothesis of stability analysis and constrained control: Lyapunov sublevel sets in LaSalle-type arguments (Khalil, *Nonlinear Systems*), Blanchini's survey of set invariance (*Automatica*, 1999), control-invariant terminal sets certifying recursive feasibility in MPC (Mayne et al., *Automatica*, 2000), invariant metric balls in contraction analysis (Lohmiller and Slotine, 1998, cited in the paper), and estimated invariant regions of attraction in safe model-based RL (Berkenkamp et al., NeurIPS 2017, also cited). It plays the role that a compact state space closed under the dynamics plays in classical RL analyses. We will add these references to the remark if the reviewer finds it useful.
>
> **Why necessary: it is the learned model that needs it.** The proof controls the trajectory error through the recursion (Lemma 1) $\|\hat x_{k+1} - x_{k+1}\| \le \bar\rho\,\|\hat x_k - x_k\| + \|F(\hat x_k) - \hat F(\hat x_k)\|$, and the error quantities it consumes ($\delta$ and $\tilde\varepsilon_1$ of Assumption 5) are defined only on segments contained in $\Omega$; the recursion is therefore valid only while the **learned** rollout $\hat x_k$ remains in $\Omega$. The initial-condition-dependent invariant set that exists automatically for a contractive $F$ confines $x_k$; it says nothing about $\hat x_k$. A sharp one-dimensional example: take $F(x) = x/2$ on $\Omega = [-1, 1]$ and let $\hat F$ equal $F$ plus a smooth bump of height $h > 1/2$ near the boundary point $x = 1$. Then $\bar{\varepsilon}_\mu = \delta = 0$, yet the learned rollout started at $\hat {x} _ {0} = 1$ exits $\Omega$ at the first step, after which the stated hypotheses constrain nothing and the return gap can be made arbitrarily large. We confine the learned rollout to the region where the model error is measured is therefore unavoidable; forward invariance is the weakest natural one, and, by Lemma A, it is implied by the hypotheses that classical sup-norm analyses already make.
>
> **Change in the paper:** A remark immediately after Assumption 4 (Section 3, red) states all of the above in compact form. Lemma A continues in part 2.
>
> ## Weakness 2
>
> > "...it is not evident whether the improved invariant-measure-based bound would remain meaningful or tractable in high-dimensional settings."
>
> We added Experiment 7 (Appendix F.6, new figure) that tests exactly this, and it shows that $\bar\varepsilon_\mu$ is more tractable in high dimensions than $\varepsilon_0$, not less so. We brief the experiment 7 and the results in the following:
>
> **Setup.** For $d \in \{2, 5, 10, 20, 50, 100\}$: random stable linear closed loops with $\rho(A) = 0.8$ fixed across $d$, 8 seeds per dimension, invariant measure exact via the discrete Lyapunov equation; model error split into a dense on-support component and a thin off-support spike, so $\varepsilon_0$ is set by the spike and $\bar\varepsilon_\mu$ by the dense term.
>
> **Findings.** (1) The Monte Carlo estimation error of $\bar\varepsilon_\mu$ follows the $N^{-1/2}$ rate uniformly in $d$: the curves for $d = 2$ through $100$ overlay. (2) The looseness factor $\varepsilon_0 / \bar\varepsilon_\mu \approx 26$ is essentially constant across $d$. (3) Our bound stays within $1$-$5\times$ of the observed return gap for all $d$, while the classical bound inflates from $\approx 28\times$ to $\approx 140\times$, its prefactor carrying the non-shrinking $\varepsilon_0$ and a reward-Lipschitz constant scaling as $\sqrt d$.

---

> ### Author Response · Authors · 2026-07-07
> **Part 2: Addressing requested changes**
>
> ## Requested change #1
>
> > "The introduction part reads more like a compressed theorem statement than an intuition-building introduction. The authors are using technical control-theoretic terminology before giving intuition, which makes the first two paragraphs too dense."
>
> We agree and rewrote the opening. The new first paragraph contains no symbols; the classical bound and its failure modes come second; formal quantities first appear in the third paragraph. The new opening reads, in full:
>
> > "When an agent plans with a learned model of its environment, a basic question is how wrong the resulting evaluation can be when the model is wrong. If the learned dynamics differ from the true dynamics by some amount, how far can the predicted return of a fixed policy diverge from its true return? The classical answer, going back to the simulation lemma and its Lipschitz refinements, charges the worst case: It measures model error by its largest value anywhere in the state space and amplifies that error by how fast nearby trajectories can spread apart. For systems that are being actively stabilized, a robot holding its balance, a vehicle tracking a path, a controller regulating a setpoint, this worst-case accounting is badly pessimistic. The closed-loop trajectories visit only a small, stable region of the state space, so a model that is wildly inaccurate where the system never goes is penalized exactly as heavily as one that is wrong where it matters. Thus, for fixed-policy evaluation under a contracting closed loop, the return gap is controlled by the model's *average* error over the states the system actually visits, not by its worst-case error."
>
> ## Requested change #2
>
> > "There is a typo in Theorem 6: Theorem 1-5 -> Assumption 1-5."
>
> Thank you; this exposed a mechanical root cause. All theorem-like environments shared one counter, so the five assumptions consumed numbers 1-5, the main theorem rendered "Theorem 6", and assumption references fell back to the parent name, producing "Assume Theorem 1 to 5".
>
> ## Requested change #3
>
> > "...it is unclear what quantity the authors are ultimately trying to transform and why the decomposition in equation (8) is the natural object to consider."
>
> **Answer.** The telescoped return gap is controlled by $\frac{\gamma L_r}{1 - \gamma\bar\rho} \sum_{k \ge 0} \gamma^k \psi(\hat x_k)$, the one-step error evaluated *along the learned rollout*; bounding each $\psi(\hat x_k)$ by $\varepsilon_0$ at this point reproduces the classical bound (Corollary 3). The rest of the proof replaces this trajectory evaluation by $\mathbb E_\mu \psi = \bar\varepsilon_\mu$, and Equation (8) is the natural vehicle because it writes $\psi(\hat x_k)$ as the target plus the only two corrections there are:
>
> $$\psi(\hat x\_k) = \underbrace{[\psi(\hat x\_k) - \psi(x\_k)]}\_{\text{(I) model-drift}} + \underbrace{[\psi(x\_k) - \mathbb E\_\mu \psi]}\_{\text{(II) mixing transient}} + \underbrace{\mathbb E\_\mu \psi}\_{\text{(III)} = \bar\varepsilon\_\mu},$$
>
> (I) The learned rollout is the wrong trajectory (bounded via Lemmas 1-2), (II) the right trajectory is not yet mixed into $\mu$ (Kantorovich-Rubinstein, Lemma 3), (III) is the target; contraction makes (I) and (II) decay geometrically.
>
> **Change in the paper:** the subsection is retitled "Proof sketch of Theorem 1", opens with "We outline the argument here; the complete proof, with all constants tracked, is given in Appendix A", and is preceded by a new motivating paragraph saying exactly the above (Sec. 4, red); the three terms are now labeled directly in the equation.
>
> ## Requested change #4
>
> > "The experiments seem to have stochastic noise. However, Figure 1 and 2 do not report standard deviations, confidence intervals, or error bars."
>
> Both figures now quantify uncertainty, matched to the stochasticity of each panel.
>
> **Change in the paper:** Fig. 1(a): $\pm 1$ SE bands over 12 Monte Carlo batches on the return gap; $\varepsilon_0, \bar\varepsilon_\mu$ are analytic (grid integration), stated in the caption. Fig. 1(b): deterministic (grid supremum/trajectory maximum); the caption states this, rather than displaying a fabricated band. Fig. 1(c): median with interquartile bands over 8-12 seeds per $N$. Fig. 2: $\pm 1$ SE band on the observed gap; the bound curves are deterministic, stated in the caption. The bands in 1(a) and 2 are narrow because the paired (common-random-numbers) estimator is genuinely low-variance.

---

### Review · Reviewer_B4oq · 2026-06-01

**Summary Of Contributions:**

The submitted work addresses the error that bounds the model-based rollout when evaluating a fixed policy on a data-driven model a closed-loop. It analyses a common RL setting where the usual one-step model error yields a pessimistic bound for multi-step errors. This paper presents a tighter error bound under the name _Invariant-measure return-gap bound_. Under the stated assumptions, any closed-loop return is bounded for all points in a bounded operating region by a three-term bound: the expected one-step model error, the mixing transient and the model-drift transient. The paper sketches the proof of in the main body, discusses consequences and reports some numerical experiments.

**Additional Comments:**

The following works might be interesting as related work for this script:
- Why long model-based rollouts are no reason for bad Q-value estimates, ESANN 2024, Wissmann, Hein, Udluft, Tresp. I think this publication could be interesting as a motivating example for the script. The stated publication gives some empirical insights in how the pessimistic one-step model error leads to an unjust bias against model-based rollouts with a fixed policy in a closed-loop setting.
- Model Regularization for Stable Sample Rollouts, Uncertainty in Artificial Intelligence 2014, Talvitie. The following holds in my opinion also for subsequent publications. I think the publications (and/ or its successors) could be potentially relevant because they address the consequences of the one-step model error on model-based RL algorithms.

**Audience:**

Yes

**Audience Explanation:**

The findings of the paper would be highly relevant to model-based reinforcement learning methods in general.
For a certain subclass of RL problems, namely problems whose good policies exhibit contracting behaviors, the presented bound sheds valuable insights on the viability of model-based methods.

**Broader Impact Concerns:**

I have no such concerns.

**Claims And Evidence:**

No

**Claims Explanation:**

The paper's main contribution is mathematical (main theorem in the paper) and, thus, supported by a sketched proof in the main body and a full proof in the appendix. If I did not overlook major errors, the stated main contribution would be supported--if several major shortcomings would be overlooked. In its current state, the paper seems to be in a _first draft_ state rather than a _submission ready_ state. There are inconsistencies, incomplete sentences, missing notations, as well as missing explanations, wrong labels and cross-link problems. Since the main contribution is mathematical, the script's main benefit would be a comprehensible presentation of its main theorem. This is not given in the current state of the paper. See "Requested Changes".

**Requested Changes:**

The overall comprehensibility of the script leaves room for improvement.

The general organisation and the central thread are difficult to follow. One key challenge is that the narrative order and the order in the document clinch.

In the following, I will try to give representative, but not exhaustive, concrete examples that should be improved:
- In the introduction, Main focus, the announced order is: main theorem -> the bridge from deterministic to stochastic -> experiments -> theoretical consequences; compared to where it’s found: section 4 -> appendix -> section 5 -> section 4.
- Assumptions, lemmas and theorems are all cited as theorems, e.g., the main theorem which is also the first stated theorem is numbered “Theorem 6”, assumptions are referred to as “Assume Theorem 1 to 5.”.
- For readability, independent numbering could really improve the script, or alternatively using subsubsection structure as is used in some mathematical scripts.
- Equations should be referenced as, e.g., “Equation 3” instead of “equation 3”, similar with “Panel (a)” vs. “panel a”.
Complete sentences after “:” should start uppercased.
- Not all referred notation is properly introduced, e.g., in Equation 8, “Term (III)” is referenced, but never introduced, Rademacher complexity has an “?” Rendered, Equation 11 uses an undefined cdot, some inline equations in the sketched proof lack limits.
- The script uses unclean abbreviated notation that goes beyond what should be expected from the TMLR review, e.g., referring to “(Tonelli, non-negative terms)” as part of a sketched proof instead of referring to “(Tonelli’s theorem)” or maybe “(see Tonelli’s theorem since the terms are non-negative)”.
- The included figures use inconsistent title structures, as well as inconsistent pointers (*(a)* vs. *Left:* vs. *Left,*).
- The figures have multiple inconsistencies, Figure 1 (c) has two lines plotted with barely any contrast to the grid, the legend in Figure 2 left depicts a red line with circles which is not plotted, also references the main theorem by its subsection number, Figure 2 right has an overlap between legend and the textbox in red.
- The push the proof into the appendix and only use a sketched is in my opinion a good idea, it should also be communicated clearly in the script.

---

> ### Author Response · Authors · 2026-07-07
> **Part I**
>
> We thank the reviewer for the review. We agree that the presentation did not meet the standard required for a primarily mathematical contribution. Thus, we treated the review as a checklist, fixed every item, and performed a full proofreading pass beyond the listed examples. A revised PDF is uploaded with changes marked in color
>
> ## Requested change #1
> > "One key challenge is that the narrative order and the order in the document clinch. ... the announced order is: main theorem -> the bridge from deterministic to stochastic -> experiments -> theoretical consequences; compared to where it's found: section 4 -> appendix -> section 5 -> section 4."
>
> Answer: The "Main focus" paragraph is rewritten so the announced order matches the document: theorem and its direct consequence (Section 4) $\to$ experiments (Section 5) $\to$ stochastic extension and complete proofs (appendix).
>
> ## Requested change #2
> > "For readability, independent numbering could really improve the script, or alternatively, using subsubsection structure as is used in some mathematical scripts."
>
> We adopted the recommendations. The paper after the fix: Assumptions 1-5, Lemmas 1-3, Corollaries 1-3, main result **Theorem 1** (invariant-measure return-gap bound), hypothesis "Assume Assumptions 1 to 5".
>
> ## Requested change #3
>
> > "The push the proof into the appendix and only use a sketched is in my opinion a good idea, it should also be communicated clearly in the script."
>
> We make changes in the following: the "Main focus" paragraph ends "...complete proofs of all results appear in the appendix, with only proof sketches in the body"; the subsection is retitled "Proof sketch of Theorem 1"; and it opens with "We outline the argument here; the complete proof, with all constants tracked, is given in Appendix A."
>
> ## Requested change #4
>
> > "Equations should be referenced as, e.g., 'Equation 3' instead of 'equation 3', similar with 'Panel (a)' vs. 'panel a'." / "Complete sentences after ':' should start uppercased."
>
> We adapt the manuscript as suggested; panel references are "Panel (a)" throughout Section 5; a global pass capitalized complete sentences after colons (12 instances), leaving list-like continuations lowercase per standard usage.
>
> ## Requested change #5
>
> > "There are inconsistencies, incomplete sentences, missing notations, as well as missing explanations, wrong labels and cross-link problems."
>
> We made the major update for this request.
>
> ## Requested change #6
>
> > "in Equation 8, 'Term (III)' is referenced, but never introduced"
>
> We corrected this in the current revision
>
> ## Requested change #7
>
> > "Rademacher complexity has an '?' Rendered"
>
> The Bartlett and Mendelson (2002) entry ("Rademacher and Gaussian complexities", JMLR 3:463-482) is added.
>
> ## Requested change #8
>
> > "Equation 11 uses an undefined cdot"
>
> The dot served no purpose and is removed: the corollary now reads $|J(F) - J(\hat F)| \le (1 + o(1))\, \frac{\gamma L_r}{(1-\gamma)(1-\bar\rho)}\, \bar\varepsilon_\mu$, consistent with notation elsewhere.
>
> ## Requested change #9
>
> > "some inline equations in the sketched proof lack limits"
>
> Both geometric sums now carry explicit limits, $\sum_{k \ge 0} \gamma^k = \frac{1}{1-\gamma}$ and $\sum_{k \ge 0} (\gamma\alpha)^k = \frac{1}{1-\gamma\alpha}$, and the telescoping display carries its full index ranges, $\sum_{t=1}^{\infty} \gamma^t \sum_{k=0}^{t-1} \bar\rho^{\,t-1-k} \|\Delta_k\|$.
>
> ## Requested change #10
>
> > "referring to '(Tonelli, non-negative terms)' ... instead of referring to '(Tonelli's theorem)' or maybe '(see Tonelli's theorem since the terms are non-negative)'."
>
> The sentence now reads: "where the last step swaps the order of summation (by Tonelli's theorem, which applies since all terms are non-negative)."

---

> ### Author Response · Authors · 2026-07-07
> **Part 2**
>
> ## Requested change #11
>
> > "Figure 1 (c) has two lines plotted with barely any contrast to the grid, the legend in Figure 2 left depicts a red line with circles which is not plotted, also references the main theorem by its subsection number, Figure 2 right has an overlap between legend and the textbox in red." / "inconsistent title structures, as well as inconsistent pointers ((a) vs. Left: vs. Left,)."
>
> We corrected Figure 1 accordingly.
>
> ##  Additional comments
>
> > "Why long model-based rollouts are no reason for bad Q-value estimates, ESANN 2024, Wissmann, Hein, Udluft, Tresp. I think this publication could be interesting as a motivating example..." / "Model Regularization for Stable Sample Rollouts, UAI 2014, Talvitie. ... (and/or its successors) could be potentially relevant..."
>
> They have been added to Section 2 with substantive positioning, together with the self-correcting-models successor (Talvitie, AAAI 2017), per the "and/or its successors" remark. The added passage reads, in full:
>
> > "Empirically, Wissmann et al. (2024) observe exactly the phenomenon our bound formalizes: For fixed-policy closed-loop evaluation, long model-based rollouts yield far better value estimates than worst-case one-step-error reasoning predicts, so the pessimism of the sup-norm accounting is an artifact of the analysis rather than of the rollouts.  On the algorithmic side, Talvitie (2014) and the follow-up self-correcting-model line (Talvitie, 2017) address the consequences of one-step model error for rollout stability by regularizing the model toward self-consistency; our analysis is complementary, characterizing when the unregularized rollout error is already benign because the closed loop contracts."
>
> In our notation: Wissmann et al. document that $\varepsilon_0$-based reasoning unjustly penalizes fixed-policy closed-loop rollouts, precisely the $\varepsilon_0$ versus $\bar\varepsilon_\mu$ gap that Theorem 1 quantifies; Talvitie makes rollout error benign *by construction*, our theorem characterizes when it is benign *already* ($\bar\rho < 1$).
>
> > "Since the main contribution is mathematical, the script's main benefit would be a comprehensible presentation of its main theorem. This is not given in the current state of the paper."
>
> We hope the revision resolves that concern, for the following reasons: every listed item is fixed; the two dominant error classes had single mechanical causes that are now removed at the source, so the fix is class-wide rather than instance-wise. We also made the substantive additions, a new high-dimensional experiment (Experiment 7, added per Reviewer 1's request) and a short remark after Assumption 4, both marked in red.

---

### Review · Reviewer_MpTt · 2026-07-16

**Summary Of Contributions:**

This paper claims to sharpen the classical bound on the discounted return gap between a fixed policy evaluated on a true dynamical
system and on a learned closed-loop model, using the invariant measure. It is a mathematically rigorous paper with empirical results supporting the theoretical bounds. I am not an expert in RL theory so I cannot comment too much on whether this bound is a significant improvement or not.

After reading the paper, I have one question regarding the inequality (Equation 3) proven in Theorem 1, which is the central result: According to Equation 7, we have

$$|J(F) - J(\hat{F})| \leq \frac{\gamma L_r}{1 - \gamma \bar{\rho}} \sum_{k} \gamma^k |\Delta_k| $$

and applying Equation 5 to it yields

$$|J(F) - J(\hat{F})| \leq \frac{\gamma L_r}{1 - \gamma \bar{\rho}} \sum_{k} \gamma^k (\delta + \tilde{\epsilon_1} |\hat{x_k} - x_* | ) \leq \frac{\gamma L_r}{1 - \gamma \bar{\rho}} \frac{\delta + \tilde{\epsilon_1} \bar{R}}{1 - \gamma}$$

using the fact that $\bar{R} = \sup_k |\hat{x_k} - x_* |$.

Note that the last term in Equation 3, namely model-drift transient, is

$$\frac{\gamma L_r}{(1 - \bar{\rho})^2} \frac{L_\psi (\delta + \tilde{\epsilon_1} \bar{R})}{1 - \gamma} \geq \frac{\gamma L_r}{(1 - \bar{\rho})} \frac{L_\psi (\delta + \tilde{\epsilon_1} \bar{R})}{1 - \gamma} \geq \frac{\gamma L_r}{1 - \gamma \bar{\rho}} \frac{L_\psi (\delta + \tilde{\epsilon_1} \bar{R})}{1 - \gamma}$$

which is greater than the bound I just derived when $L_\psi > 1$. Therefore, it seems that we can toss the first two terms in Equation 3 but still make the inequality valid, and my question is: Could the authors explain why we need to introduce the invariant measure which leads to the first term $\bar{\epsilon_\mu}$ in the bound proven in Theorem 1, given the fact that the first two terms may be redundant.

**Audience:**

Yes

**Audience Explanation:**

Please see above.

**Broader Impact Concerns:**

N/A.

**Claims And Evidence:**

No

**Claims Explanation:**

Please see above. I would be happy to change my recommendation if the authors can address my question.

**Requested Changes:**

Please see above. I hope the authors can either explain the necessity of introducing the invariant measure, or demonstrate that it is highly likely that the coefficient $L_\psi$ is sufficiently small so that my argument does not apply to the general case.

---

> ### Author Response · Authors · 2026-07-17
> **Response to Reviewer MpTt (Part 1/3): the derivation is correct; the exact comparison**
>
> We thank the reviewer for these very helpful comments. Please see below the detailed answers, which consist of 3 parts: Part 2 demonstrates that $L_\psi$ is small wherever the rollout bounds are informative; Part 3 explains why the invariant measure is needed.
>
> > "Therefore, it seems that we can toss the first two terms in Equation 3 but still make the inequality valid, and my question is: Could the authors explain why we need to introduce the invariant measure which leads to the first term $\bar\epsilon_\mu$ in the bound proven in Theorem 1, given the fact that the first two terms may be redundant."
>
> **Answer.**  The derivation is correct, and in the regime $L_\psi \gtrsim 1 - \bar\rho$, the reviewer is right: the first two terms of Equation 3 can be tossed, and the resulting bound is both valid and tighter.
> We have added that bound to the revised paper as Corollary 4 (tube-supremum baseline), credited to this review.
>
> **To be precise about scope**: the reviewer's comment does not affect the validity of Theorem 1, and the derived bound is not uniformly tighter. It is a second valid bound, obtained from two steps of our own proof (Equation 7 plus Lemma 2), tighter than Equation 3 in one region of parameter space and looser in another; two correct upper bounds on the same quantity coexist, and the substantive question, answered exactly below, is which is smaller where. The invariant measure is what charges the rollout for the error where it spends its time rather than the worst error it ever meets: the reviewer's bound charges the tube supremum $\delta + \tilde\varepsilon_1\bar R$ at the full $\frac{1}{1-\gamma}$ rate, whereas Theorem 1 charges $\bar\varepsilon_\mu$ (equal to the anchor error $\delta$ in the deterministic equilibrium case) at that rate and relegates the tube supremum to a transient that is *second order* in the model error. When the model is Jacobian-accurate, $L_\psi \le \sup_\Omega\|\nabla F - \nabla\hat F\| \ll 1 - \bar\rho$ (now Remark 1), Theorem 1 is tighter by up to $(1-\bar\rho)/L_\psi$, or $1 + \tilde\varepsilon_1\bar R/\delta$ as $L_\psi \to 0$. In the stochastic and limit-cycle settings the first two terms cannot be removed at all (no pathwise tube bound exists there, and the leading term is irreducibly $\mathbb E_\mu\psi$), and $\bar\varepsilon_\mu$ is the only quantity in any of these bounds that is estimable from rollout data (Corollary 1).
>
> **The reviewer's bound (now Corollary 4, Equation 13).** Combining Equation 7 with Lemma 2 (Equation 5) and summing the geometric series gives, under Assumptions 1-5 alone (no mixing hypothesis, no Lipschitz-$\psi$ hypothesis):
>
> $$\bigl\lvert J(F) - J(\hat F) \bigr\rvert \\le\ \frac{\gamma L_r\,(\delta + \tilde\varepsilon_1 \bar R)}{(1 - \gamma\bar\rho)(1 - \gamma)}.$$
>
> This is a *tube-supremum baseline*: it charges the largest value of the error envelope over the states the learned rollout visits, for every one of the $\frac{1}{1-\gamma}$ effective steps. It is the natural intermediate rung of a ladder the revised paper now states explicitly: the classical bound charges $\varepsilon_0 = \sup_\Omega \psi$ (Corollary 3), the reviewer's bound charges the tube envelope $\delta + \tilde\varepsilon_1 \bar R \ge \sup_{k} \psi(\hat x_k)$, and Theorem 1 charges the measure average $\bar\varepsilon_\mu = \mathbb E_\mu \psi$ plus transients.
>
> **The exact comparison.** In the case the reviewer analyzed (deterministic equilibrium, where $\mu = \delta_{x_\star}$ and hence $\bar\varepsilon_\mu = \delta$; the paper notes after Theorem 1 that the deterministic bound is "in the literal-equilibrium sense, an anchor-error theorem"), take $\gamma \to 1$ so that the mixing transient vanishes and both prefactors coincide. Then
>
> $$\text{Theorem 1} \le \text{reviewer's bound} \quad\Longleftrightarrow\quad L_\psi \le (1 - \bar\rho)\frac{\tilde\varepsilon_1 \bar R}{\delta + \tilde\varepsilon_1 \bar R},$$
>
> and when the condition holds, the improvement factor is exactly $\frac{\delta + \tilde\varepsilon_1 \bar R}{\delta + L_\psi(\delta + \tilde\varepsilon_1 \bar R)/(1 - \bar\rho)}$, with the two clean limits $(1-\bar\rho)/L_\psi$ (as $\delta \to 0$) and $1 + \tilde\varepsilon_1 \bar R / \delta$ (as $L_\psi \to 0$); at $\delta = 0$ and finite $\gamma$ the factor is $\big[\frac{1-\gamma}{1-\bar\rho} + \frac{L_\psi(1-\gamma\bar\rho)}{(1-\bar\rho)^2}\big]^{-1} \approx \frac{1-\bar\rho}{\max(1-\gamma,\, L_\psi)}$. When the condition fails, i.e. $L_\psi \gtrsim 1 - \bar\rho$, the reviewer is right: the first two terms of Equation 3 are redundant and the reviewer's bound is the tighter one. This is exactly the complement of the regime condition (Equation 10) under which Corollary 2 states the paper's headline claim; the reviewer has identified the other side of that boundary, and the revised paper now states the comparison explicitly (the paragraph after Corollary 4, with the crossover displayed as Equation 14) rather than leaving it implicit in Corollary 2.

---

> ### Author Response · Authors · 2026-07-17
> **Response to Reviewer MpTt (Part 2/3): L_psi is small wherever rollout bounds are informative; exact numbers**
>
> > "I hope the authors can either explain the necessity of introducing the invariant measure, or demonstrate that it is highly likely that the coefficient $L_\psi$ is sufficiently small so that my argument does not apply to the general case."
>
> Either alternative would settle the question; we provide both. This part demonstrates that $L_\psi$ is small in every regime where any rollout bound is informative; part 3 explains the necessity of the invariant measure.
>
> **$L_\psi$ is structurally small: three independent demonstrations.**
>
> **(a) $L_\psi$ is a first-order model-error quantity.** For $F, \hat F \in C^1$ on the convex $\Omega$, the reverse triangle inequality gives $|\psi(x) - \psi(y)| \le \lVert (F - \hat F)(x) - (F - \hat F)(y)\rVert \le \sup_{\xi \in \Omega}\lVert \nabla F(\xi) - \nabla \hat F(\xi)\rVert \lVert x - y\rVert$, hence
>
> $$L_\psi \le \sup_{\Omega} \bigl\lVert \nabla F - \nabla \hat F\bigr\rVert.$$
>
> So $L_\psi > 1$ means the learned model's Jacobian is wrong by more than $1$ in operator norm somewhere on $\Omega$, i.e. by more than the entire contraction budget ($1 - \bar\rho < 1$): a model whose local sensitivities are off by that much gives rollouts with no meaningful error control under any analysis in this family. We state this as a $C^1$ fact deliberately: sup-norm accuracy alone does *not* make $L_\psi$ small (a small but rapidly oscillating error field has $\varepsilon_0 \to 0$ with $L_\psi$ arbitrarily large), which is exactly why hypothesis (b) is a separate condition and why the comparison genuinely has two regimes. Under Jacobian-accurate learning, the standard situation for smooth model classes and the situation Assumption 5 quantifies, $L_\psi$ is first order in the model error, making the drift transient a product of two first-order quantities. This is now Remark 1 in the revised paper, placed directly after Theorem 1.
>
> **(b) The stochastic theorem already excludes the large-$L_\psi$ regime by hypothesis.** The stochastic extension (Theorem in Appendix B) requires $\gamma L_\psi < 1 - \gamma\beta$, i.e. $L_\psi < (1-\gamma\beta)/\gamma$, which is $< 1$ for all $\gamma > 1/(1+\beta)$; the simpler sufficient form used there is $L_\psi < 1 - \beta$. So in the setting where the invariant measure is non-degenerate, $L_\psi \ge 1$ is outside the theorem's hypotheses entirely.
>
> **(c) In the finite-sample experiment, $L_\psi \to 0$ at the statistical rate.** In Experiment 3 (LQR identification), the closed-loop error field is linear, $\psi(x) = \lVert M x\rVert$ with $M$ the parameter-identification error, so $L_\psi = \tilde\varepsilon_1 = \lVert M\rVert_2$ *exactly*, and $\lVert M\rVert_2 = O(\sigma/\sqrt N)$. $L_\psi$ is not merely "likely small": it decays to zero at the estimation rate, and the crossover condition $L_\psi \le 1 - \bar\rho$ is met for all $N$ beyond a fixed threshold.
>
> **Exact numbers.** Because the deterministic linear case admits closed forms for every quantity in both bounds ($\delta = 0$, $L_\psi = \tilde\varepsilon_1 = \lVert M\rVert$, $\bar R = \lVert x_0\rVert$, $C_{\mathrm{mix}} = \bar R$, $\alpha = \bar\rho$), we computed the comparison exactly: $F(x) = Ax$ with $\bar\rho = \lVert A\rVert = 0.9$, $\hat F = (A + M)x$, $\lVert x_0\rVert = 1$, $L_r = 2$, reward $-\lVert x\rVert^2$. First, sweeping the error size at $\gamma = 0.99$ (columns: exact gap; full three-term Theorem 1; reviewer's bound; ratio reviewer/ours):
>
> | $L_\psi = \lVert M\rVert$ | gap | Thm 1 (full) | reviewer | ratio |
> |---|---|---|---|---|
> | 0.001 | 0.0001 | 0.20 | 1.82 | 9.0 |
> | 0.01 | 0.006 | 3.80 | 18.2 | 4.8 |
> | 0.05 | 0.13 | 58.6 | 90.8 | 1.6 |
> | 0.1 | 0.36 | 216 | 182 | 0.84 |
> | 0.2 | 0.44 | 828 | 363 | 0.44 |
>
> The crossover sits between $\lVert M\rVert = 0.05$ and $0.1$, matching the predicted threshold $1 - \bar\rho = 0.1$. Below it, Theorem 1 is tighter; above it, the reviewer's bound is tighter, exactly as the reviewer argued. Second, fixing $L_\psi = 0.01$ and sweeping the horizon:
>
> | $\gamma$ | gap | Thm 1 (full) | reviewer | ratio |
> |---|---|---|---|---|
> | 0.9 | 0.0014 | 1.13 | 0.95 | 0.84 |
> | 0.99 | 0.0059 | 3.80 | 18.2 | 4.8 |
> | 0.999 | 0.0069 | 22.0 | 198 | 9.0 |
>
> The true gap *saturates* in the horizon (a contracting deterministic rollout pays only a transient price), and the reviewer's bound structurally cannot reproduce this: it is first order in model error times $\frac{1}{1-\gamma}$, so it grows without bound, while Theorem 1's horizon-growing part is second order ($L_\psi \tilde\varepsilon_1 \bar R$), giving the ratio $\to (1-\bar\rho)/L_\psi$. We report the $\gamma = 0.9$ row deliberately: at short horizons the reviewer's bound is the tighter one there too, and the comparison we add to the paper will say so. Neither bound dominates; the regimes are exactly characterized, and the good-model long-horizon regime belongs to Theorem 1.

---

> > ### Author Response · Authors · 2026-07-17
> > **Response to Reviewer MpTt (Part 3/3): why the invariant measure is not removable & changes to the paper**
> >
> > > "I hope the authors can either explain the necessity of introducing the invariant measure, or..."
> >
> > **The necessity of the invariant measure.** This helps promoting the tube-supremum bound from baseline to headline would forfeit precisely the localization the paper exists to provide, replacing the error where trajectories live with the error at the worst state they ever visit. Three concrete reasons, in increasing order of importance.
> >
> > **(i) In the deterministic equilibrium case, the invariant measure costs nothing and buys the anchor separation.** There $\mu = \delta_{x_\star}$, so "introducing $\mu$" adds no hypothesis beyond contraction, and $\bar\varepsilon_\mu = \delta$: the leading term charges the error *at the equilibrium the rollout converges to*, instead of the tube envelope $\delta + \tilde\varepsilon_1 \bar R$. The improvement factor $1 + \tilde\varepsilon_1 \bar R/\delta$ from part 1 is unbounded as $\delta \to 0$; a model that is exact where the system settles but imperfect along the approach has zero leading-order gap under Theorem 1, which no bound built from the tube envelope can express.
> >
> > **(ii) In the stochastic and limit-cycle settings the reviewer's route does not exist.** There $\mu$ is non-degenerate, trajectories are random, and Lemma 2's envelope evaluated along realized states has no pathwise supremum controlled by the assumptions; the stochastic proof instead absorbs the drift into a self-bounding factor (requiring $L_\psi < (1-\gamma\beta)/\gamma$) and the leading term is irreducibly $\mathbb E_\mu \psi$, with no $(\delta + \tilde\varepsilon_1 \bar R)$ term to fall back on. The paper's non-degenerate experiments live here: in Experiment 7 the measure-averaged bound stays within $1$-$5\times$ of the observed gap up to $d = 100$ while the supremum-based accounting inflates $28$-$140\times$.
> >
> > **(iii) $\bar\varepsilon_\mu$ is the only operationally estimable handle.** Corollary 1 converts Theorem 1 into a statement about the on-policy training MSE, estimable from rollout data at the $N^{-1/2}$ rate uniformly in dimension (Experiment 7). The reviewer's bound is built from $\delta$, $\tilde\varepsilon_1$, $\bar R$, which are anchor and Jacobian-supremum quantities: certifying $\tilde\varepsilon_1$ from point evaluations inherits the same exponential-in-dimension packing obstruction as certifying $\varepsilon_0$. Even in regimes where the two bounds are numerically comparable, only the invariant-measure form converts into a guarantee about the model one actually trains.
> >
> > **Changes made in the revised paper** (all marked in red; the revised PDF is uploaded):
> >
> > 1. **Corollary 4 (tube-supremum baseline, Equation 13), credited to this review**: the reviewer's bound with its one-line proof (Equation 7 + Lemma 2 + geometric sum), placed with Corollaries 1-3.
> > 2. **A comparison paragraph ("Where each bound is tighter")** stating the ladder $\varepsilon_0 \to (\delta + \tilde\varepsilon_1\bar R) \to \bar\varepsilon_\mu$, the exact crossover as Equation 14, the improvement factor and its two limits, and the explicit statement that above the threshold the tube baseline is the tighter bound and the transients of Equation 3 are redundant; cross-referenced to Corollary 2's condition (Equation 10), which is its complement.
> > 3. **Remark 1**: the display $L_\psi \le \sup_\Omega\|\nabla F - \nabla\hat F\|$ with its one-line justification and the supremum-versus-Jacobian caveat, directly after Theorem 1, so that $L_\psi$ is visibly a first-order model-error quantity rather than a free constant.
> > 4. **A precision fix to Experiment 4**: The bound curve in Figure 2 represents only the leading term of Equation 3, but this was not explicitly stated in the original manuscript. The revised legend, caption, and body text now identify it accordingly. Appendix F.4 also reports the closed-form transient magnitudes for this construction: $\delta \approx 0.066c$; $\tilde{\varepsilon}1 = L\psi \approx 19.4c$, arising from the deliberately sharp bump; a mixing term of approximately $20c$; a drift term of approximately $8.1\times10^3 c^2$; and a tube baseline of approximately $270c$. These revisions clearly distinguish the quantity plotted in Figure 2 from the full right-hand side of Equation 3.

---

### Author Response · Authors · 2026-07-17
**REVISION NOTE**

We thank three reviewers for their careful and concrete reviews. We have uploaded a revised PDF in which all substantive changes are marked in red.

**Changes at a glance.**

*New content*
- Experiment 7 (Appendix F.6): high-dimensional tractability study, state dimension up to $d = 100$, testing whether the invariant-measure-averaged error $\bar\varepsilon_\mu$ remains estimable as dimension grows (Reviewer Eb3m, Weakness 2).

- A remark after Assumption 4 explaining why forward invariance is required for the learned model $\hat F$, not only the true system, and why the assumption is mild (Reviewer Eb3m, Weakness 1; a formal lemma with proof is stated in our response).

- Related work: Wissmann et al. (ESANN 2024) added as empirical motivation, and Talvitie (UAI 2014) with the follow-up self-correcting-model line (AAAI 2017) as the complementary algorithmic response to one-step model error (Reviewer B4oq, suggestions).

*Structural*
- Independent numbering for all theorem-like environments: the main theorem is now Theorem 1 (previously rendered "Theorem 6" because assumptions shared its counter), the assumptions are Assumptions 1-5, and the hypothesis reads "Assume Assumptions 1 to 5"

- The introduction's opening is rewritten (Reviewer Eb3m), and its "Main focus" paragraph is rewritten so the announced order of results matches the document order (Reviewer B4oq).

- The proof subsection is retitled "Proof sketch of Theorem 1", opens with an explicit pointer to the complete proof in Appendix A, and is preceded by a new paragraph motivating the decomposition (both reviewers).

*Presentation and figures*

- Error bars or explicitly stated determinism for every panel of Figures 1 and 2; five specific figure defects fixed; all figures unified to per-panel (a)/(b)/(c) titles with matching caption pointers.

- Notation and rendering fixes: Term (III) labeled, the unresolved Rademacher citation fixed, explicit summation limits, a stray multiplication dot removed, and the abbreviated Tonelli reference written out.

Additionally, we include the following,

1. **Corollary 4 (tube-supremum baseline, Equation 13), credited to this review**: the reviewer's bound with its one-line proof (Equation 7 + Lemma 2 + geometric sum), placed with Corollaries 1-3.

2. **A comparison paragraph ("Where each bound is tighter")** stating the ladder $\varepsilon_0 \to (\delta + \tilde\varepsilon_1\bar R) \to \bar\varepsilon_\mu$, the exact crossover as Equation 14, the improvement factor and its two limits, and the explicit statement that above the threshold the tube baseline is the tighter bound and the transients of Equation 3 are redundant; cross-referenced to Corollary 2's condition (Equation 10), which is its complement.

3. **Remark 1**: the display $L_\psi \le \sup_\Omega\|\nabla F - \nabla\hat F\|$ with its one-line justification and the supremum-versus-Jacobian caveat, directly after Theorem 1, so that $L_\psi$ is visibly a first-order model-error quantity rather than a free constant.

4. **A precision fix to Experiment 4**: The bound curve in Figure 2 represents only the leading term of Equation 3, but this was not explicitly stated in the original manuscript. The revised legend, caption, and body text now identify it accordingly. Appendix F.4 also reports the closed-form transient magnitudes for this construction: $\delta \approx 0.066c$; $\tilde{\varepsilon}1 = L\psi \approx 19.4c$, arising from the deliberately sharp bump; a mixing term of approximately $20c$; a drift term of approximately $8.1\times10^3 c^2$; and a tube baseline of approximately $270c$. These revisions clearly distinguish the quantity plotted in Figure 2 from the full right-hand side of Equation 3.